# EXACT STOCHASTIC NEWTON METHOD FOR DEEP LEARNING: THE FEEDFORWARD NETWORKS CASE.

## ABSTRACT

The inclusion of second-order information into Deep Learning optimization has drawn consistent interest as a way forward to improve upon gradient descent methods. Estimating the second-order update is computationally expensive, which drastically limits its usage scope and forces the use of various truncations and approximations. This work demonstrates that it is possible to solve the Newton direction in the stochastic case exactly. We consider feedforward networks as a base model, build a second-order Lagrangian which we call Sifrian, and provide a closed-form formula for the exact stochastic Newton direction under some monotonicity and regularization conditions. We propose a convexity correction to escape saddle points, and we reconsider the intrinsic stochasticity of the online learning process to improve upon the formulas. We finally compare the performance of the developed solution with well-established training methods and show its viability as a training method for Deep Learning.

Optimization in Deep Learning is mainly dominated by first-order methods built around the central concept of backpropagation (LeCun et al., 1988). Second-order methods have exceptional theoretical properties in deterministic optimization (Boyd et al., 2004; Nocedal & Wright, 2006), but such properties do not translate well into the stochastic case (LeCun et al., 2012; Bottou et al., 2018). First-order methods such as the Stochastic Gradient Descent (SGD) (Robbins & Monro, 1951) are relatively simple and have been adaptively improved for Deep Learning (Duchi et al., 2011; Tieleman & Hinton, 2012; Kingma & Ba, 2014; Reddi et al., 2019; Yao et al., 2020). Inherently, second-order methods might seems inadequate for Deep learning due to increased computational cost, poor clock-wall performance, and the non-convex nature of Deep Learning (LeCun et al., 2012). Furthermore, Newton method might even reduce the generalization capabilities of training (Wadia et al., 2021; Amari et al., 2020). Despite these various limitations, substantial effort was deployed to include Hessian information into the optimization process (e.g. Byrd et al., 2011; Sohl-Dickstein et al., 2014; Byrd et al., 2016; Agarwal et al., 2017; Berahas et al., 2019; Anil et al., 2020; Goldfarb et al., 2020; Castera et al., 2021). Several approaches exist such as the Gauss-Newton method (e.g Schraudolph, 2002; Botev et al., 2017), diagonal approximation of the Hessian (e.g. Bordes et al., 2009; Schaul et al., 2013), iterative low-rank updates such as BFGS (Broyden, 1970; Fletcher, 1970; Goldfarb, 1970; Shanno, 1970) (see also, Liu & Nocedal, 1989; Schraudolph et al., 2007; Bollapragada et al., 2018), or Hessian-Free methods which combine the fast Hessian matrix multiplication (Pearlmutter, 1994) and the conjugate gradient algorithm (e.g. Martens, 2010; Martens & Sutskever, 2012; Dauphin et al., 2014). The use of the Fisher matrix to capture curvature information in the space of distributions, instead of the Hessian, is another approach which yields the natural gradient (Amari, 1998), but suffers from the same computational issues as the Newton method. Within this context K-FAC method (Martens & Grosse, 2015; Ba et al., 2016a; George et al., 2018) mitigates some of the computational issues of the natural gradient method. In essence, several drawbacks and flaws limits the adoption of second-order methods as a standard for neural networks training .

In this paper, we propose a novel approach to characterize the Newton update which help us derive an exact closed-form solution for the stochastic Newton method. Our method requires a suitable regularization of the neural network and strict monotonicity of the activation functions. We start this paper by introducing useful notations while deriving the well-known backpropagation algorithm for a feedforward network. Then we introduce a second-order Lagrangian which we call Sifrian, that will serve to characterize the Newton direction. We derive four types of equations from the Sifrian, and we provide an exact closed-form solution for the Newton direction in the stochastic case. We

further propose a saddle-free version of our method, and add a randomization process to enhance our solution. In the last part of this paper, we show the applicability of our method for Deep Learning through diverse classification tasks using feedforward architectures.

# 1 PRELIMINARIES

Deep Learning and neural networks training could be seen as an optimization problem of an expected loss function $\ell$ over a distribution $\mathcal{D}$ of labeled samples $(\mathbf{x}, \mathbf{d})$. In general, weights and biases $(\mathbf{W}, \boldsymbol{\beta})$ are the sought-after parameters of the network. The labeled samples distribution $\mathcal{D}$ is often unknown, but a large number of samples (database $\mathbb{D}$) allows the approximation of the expected loss with an empirical risk:

$$\min_{\boldsymbol{W}, \boldsymbol{\beta}} \left( \mathbb{E}_{(\mathbf{x}, \mathbf{d}) \sim \mathcal{D}} \left[ \ell \left( \boldsymbol{W}, \boldsymbol{\beta}, \mathbf{x}, \mathbf{d} \right) \right] \right) \rightarrow \frac{1}{|\mathbb{D}|} \sum_{p \in \mathbb{D}} \ell \left( \boldsymbol{W}, \boldsymbol{\beta}, \mathbf{x}^{(p)}, \mathbf{d}^{(p)} \right), \ \left( \mathbf{x}^{(p)}, \mathbf{d}^{(p)} \right)_{p \in \mathbb{D}} \overset{i.i.d}{\sim} \mathcal{D}. \quad (1)$$

The loss or cost function $\ell$ is typically a cross entropy function or an $l^2$ norm of the mismatches between the network outputs and the labels: $\ell \left( \boldsymbol{W}, \boldsymbol{\beta}, \left( \mathbf{x}^{(p)}, \mathbf{d}^{(p)} \right) \right) = \frac{1}{2} \left\| \mathbf{x}_n^{(p)} - \mathbf{d}^{(p)} \right\|_2^2$. Where $n$ indicates the output layer number. We present more in detail the architecture of a Feedforward Neural Network (FNN) hereafter.

## 1.1 NOTATIONS AND FEEDFORWARD NEURAL NETWORKS (FNN)

In this section, we recall the main details of FNNs, which will be used as a standard model for Deep Learning[1]. The notation used throughout this paper is similar to notations presented in LeCun et al. (1988). The main equation governing the FNN, a.k.a. the forward model is the following:

$$\mathbf{x}_k^{(p)} = F \left( \boldsymbol{W}_k \, \mathbf{x}_{k-1}^{(p)} + \boldsymbol{\beta}_k \right), \quad k \in [1..n], \quad p \in \mathbb{D}. \quad (2)$$

where $\mathbb{D}$ is the database, $p$ designs one single sample from the database (e.g. one single image or audio recording), $k$ is the layer number and $n$ is the total number of layers in the network. The initial input for sample $p$ is $\mathbf{x}_0^{(p)}$ (e.g. the vectorized input image data). The state variable $\mathbf{x}^{(p)}$ is transformed at each layer $k$ through a multiplication by a weight matrix $\boldsymbol{W}_k$ and an addition of a bias vector $\boldsymbol{\beta}_k$. The activation function $F$, which is typically a sigmoid or a Rectified Linear Unit (ReLU), is applied element wise on the resulting activation vector: $\mathbf{a}_k^{(p)} = \boldsymbol{W}_k \, \mathbf{x}_{k-1}^{(p)} + \boldsymbol{\beta}_k$, and serves to introduce non-linearity in the neural network.

## 1.2 THE LAGRANGIAN AND BACKPROPAGATION

The origin of backpropagation could be traced back to the early 1970s and could be derived by casting Deep Learning as a constrained optimization problem. This section is similar to LeCun et al. (1988). The main novelty in this section is the addition of a regularization term dependent on the state variable $R(\mathbf{x})$ to the cost function $\ell$: $\ell_R = \ell + R(\mathbf{x})$. The regularization has to verify the admissibility criteria which we define hereafter:

**Definition 1:** A regularization term is admissible if the second derivative of the augmented cost function w.r.t. to the state variable is separable and non-null, i.e.:

$$\forall (p, q) \in \mathbb{D}, \forall (k, m) \in [1..n] \quad \left( \frac{\partial \ell_R}{\partial \mathbf{x}_k^{(p)} \mathbf{x}_m^{(q)}} \right)_{p, q, k, m} \propto \delta_{p=q}. \quad (3)$$

An example of such regularization is the following function:

$$R(\mathbf{x}) = \frac{1}{2} \sum_{p \in \mathbb{D}} \sum_{k=1..n-1} \left\langle \mathbf{x}_k^{(p)}, \boldsymbol{\Lambda}_k^{(p)} \mathbf{x}_k^{(p)} \right\rangle \quad (4)$$

---

[1]Convolutional Neural Networks (CNN) are a type of FNN. The concepts of this paper apply also to CNN.

$\left( \mathbf{\Lambda}_k^{(p)} \right)_{p,k}$ are symmetric positive matrices. Such a choice might seem atypical since the regularization often concerns the network parameters, mainly the weights or the biases. This "admissible" regularization was introduced for the sole purpose of guaranteeing a non-null partial derivative of the cost $\ell$ w.r.t the state variable $\mathbf{x}_k^{(p)}$. This last property is essential to solve Equ. 16. We will show later that such a regularization minimizes also the norm of the gradient (Barrett & Dherin, 2020; Smith et al., 2021).

In order to derive the backpropagation algorithm we introduce the following Lagrangian as per the notation of LeCun et al. (1988):

$$\mathcal{L}(\mathbf{x}, \boldsymbol{W}, \boldsymbol{\beta}, \mathbf{b}) = \ell_R + \sum_{p \in \mathbb{D}} \sum_{k=1..n} \left\langle \mathbf{x}_k^{(p)} - F\left( \boldsymbol{W}_k \, \mathbf{x}_{k-1}^{(p)} + \boldsymbol{\beta}_k \right), \mathbf{b}_k^{(p)} \right\rangle. \tag{5}$$

The Lagrangian contains the original cost function, the admissible regularization term, and the product of the forward equation with the adjoint state vectors $\left( \mathbf{b}_k^{(p)} \right)_{p,k}$. These adjoint vectors are defined for each layer of the network and each sample of the database; however, their values are not fixed yet and will be chosen in a way that simplifies the gradient computation. If the state variable $\mathbf{x}$ verifies the forward equations, then the Lagrangian simplifies to the cost function:

$$\mathcal{L}(\mathbf{x}(\boldsymbol{W}, \boldsymbol{\beta}), \boldsymbol{W}, \boldsymbol{\beta}, \mathbf{b}) = \ell_R(\mathbf{x}(\boldsymbol{W}, \boldsymbol{\beta})). \tag{6}$$

The total derivative of the previous Lagrangian w.r.t. the weights or biases is the gradient and could be expressed in terms of partial derivatives as follows[2]:

$$\begin{cases} \frac{d\mathcal{L}(\mathbf{x}(\boldsymbol{W}, \boldsymbol{\beta}), \boldsymbol{W}, \boldsymbol{\beta}, \mathbf{b})}{d\boldsymbol{W}_k} = \frac{\partial \mathcal{L}}{\partial \boldsymbol{W}_k} + \sum_{p \in \mathbb{D}} \sum_{m=1..n} \left( \frac{d\mathbf{x}_m^{(p)}}{d\boldsymbol{W}_k} \right) \left( \frac{\partial \mathcal{L}}{\partial \mathbf{x}_m^{(p)}} \right) = \frac{d\ell_R}{d\boldsymbol{W}_k}, \\[2mm] \frac{d\mathcal{L}(\mathbf{x}(\boldsymbol{W}, \boldsymbol{\beta}), \boldsymbol{W}, \boldsymbol{\beta}, \mathbf{b})}{d\boldsymbol{\beta}_k} = \frac{\partial \mathcal{L}}{\partial \boldsymbol{\beta}_k} + \sum_{p \in \mathbb{D}} \sum_{m=1..n} \left( \frac{d\mathbf{x}_m^{(p)}}{d\boldsymbol{\beta}_k} \right) \left( \frac{\partial \mathcal{L}}{\partial \mathbf{x}_m^{(p)}} \right) = \frac{d\ell_R}{d\boldsymbol{\beta}_k}. \end{cases} \tag{7}$$

The partial derivatives of the Lagrangian w.r.t to $\{ \boldsymbol{W}_k, \boldsymbol{\beta}_k \}_{k=1..n}$ are straightforward to compute. The Fréchet derivatives $\left( \frac{d\mathbf{x}_m^{(p)}}{d\boldsymbol{W}_k}, \frac{d\mathbf{x}_m^{(p)}}{d\boldsymbol{\beta}_k} \right)_{p,k,m}$ are non-trivial to evaluate and the core idea of backpropagation is the selection of the adjoint state variables $\left( \mathbf{b}_k^{(p)} \right)_{p,k}$ which cancels the superfluous terms. Such a simplification is achievable if:

$$\left( \frac{\partial \mathcal{L}}{\partial \mathbf{x}_k^{(p)}} \right)_{p \in \mathbb{D}, \, k \in [1..n]} = 0. \tag{8}$$

Computing the previous partial derivative yields the following backpropagation equation:

$$\frac{\partial \mathcal{L}}{\partial \mathbf{x}_k^{(p)}} = \mathbf{b}_k^{(p)} - \mathbf{1}_{k=1..n-1} \boldsymbol{W}_{k+1}^T \nabla F\left( \mathbf{a}_{k+1}^{(p)} \right) \mathbf{b}_{k+1}^{(p)} + \frac{\partial \ell_R}{\partial \mathbf{x}_k^{(p)}} = 0. \tag{9}$$

$\mathbf{1}_E$ is the indicator function, and it is equal to one if the underlying condition $E$ is true and null otherwise. The nabla operator $\nabla F$ is a diagonal square matrix with an element-wise derivative of its argument (along the diagonal). The resolution of the backpropagation system could be split into a boundary condition and a backward propagation system. Further details could be found in LeCun et al. (1988). The particular choice of the adjoint state vectors $\left( \mathbf{b}_k^{(p)} \right)_{p \in D, \, k=1..n}$ yields a simple formula for the gradient of the cost function:

$$\begin{cases} \boldsymbol{G}_k = \frac{\partial \mathcal{L}}{\partial \boldsymbol{W}_k} = - \sum_p \nabla F\left( \mathbf{a}_k^{(p)} \right) \mathbf{b}_k^{(p)} \mathbf{x}_{k-1}^{(p)T}, \\[2mm] \mathbf{g}_k = \frac{\partial \mathcal{L}}{\partial \boldsymbol{\beta}_k} = - \sum_p \nabla F\left( \mathbf{a}_k^{(p)} \right) \mathbf{b}_k^{(p)}. \end{cases} \tag{10}$$

The backpropagation algorithm is fundamental for Deep Learning. It is simple and provides a straightforward solution to an otherwise tedious problem to solve. Unfortunately, developing an efficient second-order backpropagation method remains elusive. The current state-of-the-art is based on the use of the $\mathcal{R}-$operator (Pearlmutter, 1994) to compute the product of the Hessian with a given vector without explicitly calculating or storing the Hessian (Martens, 2010; Dauphin et al., 2014; Agarwal et al., 2017). In the following section, we provide a different framework for backpropagation which allows the characterization of the second-order Newton direction.

---

[2] A denominator layout notation for derivation is used throughout this paper.

## 2 THE SIFRIAN AND THE EXACT NEWTON DIRECTION

It is legitimate to wonder if the Lagrangian multipliers method could be replicated for higher-order optimization. We hereafter provide a solution for the second-order case and build an equivalent Lagrangian which we call the Sifrian.

### 2.1 THE SIFRIAN

The Lagrangian is useful to derive the first-order gradient, and it combines the cost function and the forward model multiplied by adjoint variables. Our approach to extend the Lagrangian consists in creating a function with the forward model, the backward model, and the definition of the gradient. We choose to omit the loss function, as its derivation will generate the gradient again. Starting from this paradigm, we introduce the Sifrian[3].

**Definition 2:** The Sifrian $\mathcal{S}$ of the feed-forward network is defined as follows:

$$
\mathcal{S}(\mathbf{x}, \mathbf{b}, \boldsymbol{W}, \boldsymbol{\beta}, \boldsymbol{G}, \mathbf{g}, \boldsymbol{\gamma}, \boldsymbol{\zeta}, \boldsymbol{N}, \boldsymbol{\eta}) = \sum_{k,p} \left\langle \mathbf{x}_k^{(p)} - F\left(\mathbf{a}_k^{(p)}\right), \boldsymbol{\gamma}_k^{(p)} \right\rangle
$$

$$
+ \left\langle \mathbf{b}_k^{(p)} - \mathbf{1}_{k=1..n-1} \boldsymbol{W}_{k+1}^T \nabla F\left(\mathbf{a}_{k+1}^{(p)}\right) \mathbf{b}_{k+1}^{(p)} + \frac{\partial \ell_R}{\partial \mathbf{x}_k^{(p)}}, \boldsymbol{\zeta}_k^{(p)} \right\rangle
$$

$$
+ \sum_k \left\langle \boldsymbol{G}_k + \sum_p \nabla F\left(\mathbf{a}_k^{(p)}\right) \mathbf{b}_k^{(p)} \mathbf{x}_{k-1}^{(p)T}, \boldsymbol{N}_k \right\rangle
$$

$$
+ \left\langle \mathbf{g}_k + \sum_p \nabla F\left(\mathbf{a}_k^{(p)}\right) \mathbf{b}_k^{(p)}, \boldsymbol{\eta}_k \right\rangle. \tag{11}
$$

The Sifrian includes all the equations generated by the first order backpropagation, multiplied by four new adjoint parameters $\left\{ \boldsymbol{\gamma}_k^{(p)}, \boldsymbol{\zeta}_k^{(p)}, \boldsymbol{N}_k, \boldsymbol{\eta}_k \right\}$. While the Lagrangian describes a saddle point (Nocedal & Wright, 2006), the Sifrian, describes an equilibrium and it has a unique null value when all the forward, backward and gradients equations are verified i.e.:

$$
\mathcal{S}(\mathbf{x}\left(\boldsymbol{W}, \boldsymbol{\beta}\right), \mathbf{b}\left(\boldsymbol{W}, \boldsymbol{\beta}, \mathbf{x}\right), \boldsymbol{W}, \boldsymbol{\beta}, \boldsymbol{G}\left(\boldsymbol{W}, \boldsymbol{\beta}, \mathbf{x}, \mathbf{b}\right), \mathbf{g}\left(\boldsymbol{W}, \boldsymbol{\beta}, \mathbf{x}, \mathbf{b}\right), \boldsymbol{\gamma}, \boldsymbol{\zeta}, \boldsymbol{N}, \boldsymbol{\eta}) = 0. \tag{12}
$$

We call the previous quantity the equilibrated Sifrian, and it is critical to notice its invariance to total derivation w.r.t. either the weights or biases (i.e., remains null). Expressing the total derivative of the equilibrated Sifrian using partial derivatives yields the following two equations:

$$
\begin{cases}
\frac{d\mathcal{S}}{d\boldsymbol{W}_k} = \frac{\partial\mathcal{S}}{\partial\boldsymbol{W}_k} + \sum_{m,p} \frac{d\mathbf{x}_m^{(p)}}{d\boldsymbol{W}_k} \frac{\partial\mathcal{S}}{\partial\mathbf{x}_m^{(p)}} + \frac{d\mathbf{b}_m^{(p)}}{d\boldsymbol{W}_k} \frac{\partial\mathcal{S}}{\partial\mathbf{b}_m^{(p)}} + \sum_m \frac{d\boldsymbol{G}_m}{d\boldsymbol{W}_k} \frac{\partial\mathcal{S}}{\partial\boldsymbol{G}_m} + \frac{d\mathbf{g}_m}{d\boldsymbol{W}_k} \frac{\partial\mathcal{S}}{\partial g_m} = 0, \\[2ex]
\frac{d\mathcal{S}}{d\boldsymbol{\beta}_k} = \frac{\partial\mathcal{S}}{\partial\boldsymbol{\beta}_k} + \sum_{m,p} \frac{d\mathbf{x}_m^{(p)}}{d\boldsymbol{\beta}_k} \frac{\partial\mathcal{S}}{\partial\mathbf{x}_m^{(p)}} + \frac{d\mathbf{b}_m^{(p)}}{d\boldsymbol{\beta}_k} \frac{\partial\mathcal{S}}{\partial\mathbf{b}_m^{(p)}} + \sum_m \frac{d\boldsymbol{G}_m}{d\boldsymbol{\beta}_k} \frac{\partial\mathcal{S}}{\partial\boldsymbol{G}_m} + \frac{d\mathbf{g}_m}{d\boldsymbol{\beta}_k} \frac{\partial\mathcal{S}}{\partial\mathbf{g}_m} = 0.
\end{cases} \tag{13}
$$

Similar to first order backpropagation, we select the new adjoints variables $\left\{ \boldsymbol{\gamma}_k^{(p)}, \boldsymbol{\zeta}_k^{(p)}, \boldsymbol{N}_k, \boldsymbol{\eta}_k \right\}_{p \in \mathbb{D}, \, k=1..n}$ to circumvent the calculation of any superfluous terms. We make the following choice:

$$
\forall\, m, \forall\, q : \quad \frac{\partial\mathcal{S}}{\partial\boldsymbol{W}_m} = -\boldsymbol{G}_m, \quad \frac{\partial\mathcal{S}}{\partial\boldsymbol{\beta}_m} = -\mathbf{g}_m, \quad \frac{\partial\mathcal{S}}{\partial\mathbf{x}_m^{(q)}} = 0, \quad \frac{\partial\mathcal{S}}{\partial\mathbf{b}_m^{(q)}} = 0. \tag{14}
$$

In this case, after vectorizing the weight gradient $\boldsymbol{G}_k$ (and also $\boldsymbol{N}_k$), and concatenating all their values across all layers, the Equ. 13 could be cast as the following block matrix system:

$$
\begin{bmatrix} \frac{d\boldsymbol{G}}{d\boldsymbol{W}} & \frac{d\mathbf{g}}{d\boldsymbol{W}} \\ \frac{d\boldsymbol{G}}{d\boldsymbol{\beta}} & \frac{d\mathbf{g}}{d\boldsymbol{\beta}} \end{bmatrix} \begin{bmatrix} \boldsymbol{N} \\ \boldsymbol{\eta} \end{bmatrix} = \begin{bmatrix} \boldsymbol{G} \\ \mathbf{g} \end{bmatrix} \rightarrow \boldsymbol{H} \begin{bmatrix} \boldsymbol{N} \\ \boldsymbol{\eta} \end{bmatrix} = \begin{bmatrix} \boldsymbol{G} \\ \mathbf{g} \end{bmatrix}. \tag{15}
$$

---

[3]The choice of the Sifrian appellation stems from the Arabic word "Sifr", which means zero. "Rien" means also "nothing" in French.

The matrix that appears in the previous system ($\boldsymbol{H}$) is the Hessian. Hence the computation of the Newton direction could be achieved by solving the associated system of four equations in Equ. 14. The derivation of the four types of Sifrian equations[4] is provided with further details in the Appendix (A.1). We report hereafter the equations for piece-wise affine activation functions (e.g., ReLU or Leaky ReLU). The general case of $\mathcal{C}^2$ activation function is included in the Appendix (A.1).

$$(\mathbf{1}) \sum_p \nabla F_k^{(p)} \left( \boldsymbol{\gamma}_k^{(p)} \mathbf{x}_{k-1}^{(p)T} + \mathbf{b}_k^{(p)} \mathbf{x}_{k-1}^{(p)T} + \mathbf{1}_{k=2..n} \mathbf{b}_k^{(p)} \boldsymbol{\zeta}_{k-1}^{(p)T} \right) = 0,$$

$$(\mathbf{2}) \sum_p \nabla F_k^{(p)} \left( \boldsymbol{\gamma}_k^{(p)} + \mathbf{b}_k^{(p)} \right) = 0,$$

$$(\mathbf{3}) \boldsymbol{\gamma}_k^{(p)} - \mathbf{1}_{k<n} \boldsymbol{W}_{k+1}^T \nabla F_{k+1}^{(p)} \boldsymbol{\gamma}_{k+1}^{(p)} + \sum_m \frac{\partial^2 \ell_R}{\partial \mathbf{x}_k^{(p)} \partial \mathbf{x}_m^{(p)}} \boldsymbol{\zeta}_m^{(p)} + \mathbf{1}_{k<n} \boldsymbol{N}_{k+1}^T \nabla F_{k+1}^{(p)} \mathbf{b}_{k+1}^{(p)} = 0,$$

$$(\mathbf{4}) \boldsymbol{\zeta}_k^{(p)} - \mathbf{1}_{k=2..n} \nabla F_k^{(p)} \boldsymbol{W}_k \boldsymbol{\zeta}_{k-1}^{(p)} + \nabla F_k^{(p)} \left( \boldsymbol{N}_k \mathbf{x}_{k-1}^{(p)} + \boldsymbol{\eta}_k \right) = 0.$$

$$(16)$$

**Comparison with the $\mathcal{R}$-operator:** Characterizing the Hessian effect without explicitly computing its values has been proposed by Pearlmutter (1994), using an operator $\mathcal{R}$ which requires the preselection of a specific direction to study its transformation through the Hessian. The Sifrian formulation can achieve the same goal by relaxing the two gradient constraints from the Sifrian equations and preselecting the values of $\{\boldsymbol{N}, \boldsymbol{\eta}\}$. The product requires a forward pass to compute $\boldsymbol{\zeta}$, a backward pass to compute $\boldsymbol{\gamma}$. Finally, the sought-after product is provided by the gradient of the Sifrian w.r.t. to the weights and biases. The Sifrian formulation is derived differently from the $\mathcal{R}$-operator. Nevertheless, it can still fulfill the same role, and it is more suitable for inversion, which we will conduct hereafter.

## 2.2 THE EXACT STOCHASTIC NEWTON (ESN)

The resolution of the previous system is intricate in the general case. However, if we consider a single sample, i.e., the stochastic case and a strictly monotonous activation function, then a closed-form solution could be derived.

**ESN Theorem:** If the cost function is admissibly regularized, the activation function $F$ is strictly-monotonous piece-wise affine, the first order adjoints $\left( \mathbf{b}_k^{(p)} \right)_{p,k}$ are non-null and the curvature of the output layer $\left\{ \frac{\partial^2 \ell_R}{\partial \mathbf{x}_n^{(p)} \partial \mathbf{x}_n^{(p)}} \right\}$ is invertible, then the minimal-rank stochastic Newton second-order update has a closed-form solution:

$$\begin{cases} \boldsymbol{N}_k = -\frac{\nabla F \left( \mathbf{a}_k^{(p)} \right) \mathbf{b}_k^{(p)}}{\left\| \nabla F \left( \mathbf{a}_k^{(p)} \right) \mathbf{b}_k^{(p)} \right\|_2^2} \left( \frac{\partial \ell_R}{\partial \mathbf{x}_{k-1}^{(p)}} \right)^T, \\ \\ \boldsymbol{\eta}_k = \mathbf{1}_{k=n} \left[ \nabla F \left( \mathbf{a}_n^{(p)} \right) \right]^{-1} \left\{ \frac{\partial^2 \ell_R}{\partial \mathbf{x}_n^{(p)} \partial \mathbf{x}_n^{(p)}} \right\}^{-1} \frac{\partial \ell_R}{\partial \mathbf{x}_n^{(p)}} - \boldsymbol{N}_k \mathbf{x}_{k-1}^{(p)}. \end{cases}$$

$$(17)$$

**Proof:** Under the proposed assumptions, the Sifrian second equation simplifies to $\boldsymbol{\gamma}_k^{(p)} = -\mathbf{b}_k^{(p)}$, which starts a cascade of simplifications. The first equations becomes $\mathbf{1}_{k=2..n} \boldsymbol{\zeta}_{k-1}^{(p)} = 0$. The last layer of the third equation yields: $\boldsymbol{\gamma}_n^{(p)} + \frac{\partial^2 \ell_R}{\partial \mathbf{x}_n^{(p)} \partial \mathbf{x}_n^{(p)}} \boldsymbol{\zeta}_n^{(p)} = 0$. The remaining two equations become:

$$\begin{cases} \mathbf{1}_{k=1..n-1} \frac{\partial \ell_R}{\partial \mathbf{x}_k^{(p)}} + \mathbf{1}_{k=1..n-1} \boldsymbol{N}_{k+1}^T \nabla F \left( \mathbf{a}_{k+1}^{(p)} \right) \mathbf{b}_{k+1}^{(p)} = 0, \\ \\ -\left\{ \frac{\partial^2 \ell_R}{\partial \mathbf{x}_n^{(p)} \partial \mathbf{x}_n^{(p)}} \right\}^{-1} \frac{\partial \ell_R}{\partial \mathbf{x}_n^{(p)}} + \nabla F \left( \mathbf{a}_k^{(p)} \right) \left( \boldsymbol{N}_k \mathbf{x}_{k-1}^{(p)} + \boldsymbol{\eta}_k \right) = 0. \end{cases}$$

$$(18)$$

---

[4]The Sifrian yields a total of $2 |\mathbb{D}| + 2$ equations with an equal number of unknowns.

The Newton weight is a matrix, and it is determined through its product with the adjoint vector. Inverting the Newton direction is a degenerate problem, but the minimal rank solution is unique, and it is given in Equ. 17. A major issue still remains: the results of the Newton weight do not extend to the first layer (Equ. 18). To circumvent this shortcoming, we introduce an additional layer before the input, which corresponds to the identity multiplication and a null bias. This layer remains unchanged throughout the optimization process, and we call it the "white layer". This pre-input extension allows the definition of an update rule for the first layer. □

**White layer concept** The third equation of the Sifrian allows the characterization of the Newton weight $N_k$ only between the $2^{nd}$ and the output ($n^{th}$) layers. Such a problem is **fundamental** and it is legitimate to wonder how is it possible for Hessian-Free methods to work for inversion. The answer lays within the regularization/damping (Marquardt, 1970) which covers this structural gap. The important role of the first layer for the Newton method has been highlighted recently by Wadia et al. (2021). In our case, and in order to extend the inversion result to the first layer, we just introduce an additional layer before the input, which has the same size as the input, and does not bring any modification, hence the appellation "white layer". The white layer corresponds to a matrix multiplication with the identity, a null bias, an identity activation function and is unchanged during the training.

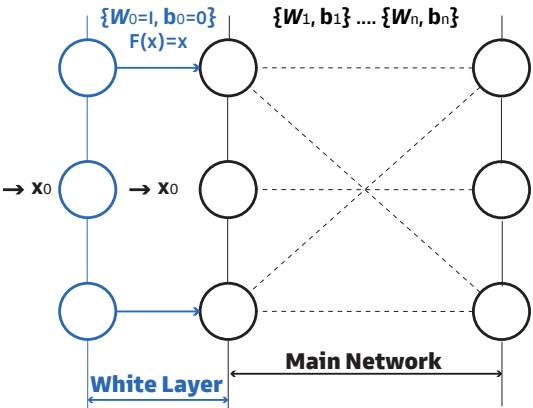

Figure 1: White layer extension before the input of the neural network.

**Admissible regularization:** The ESN solution relies primarily on the existence of the term $\left(\frac{\partial \ell_R}{\partial \mathbf{x}_k^{(p)}}\right)$ which would be null if not for the admissible regularization, whose effect revolves ultimately around minimizing the norm of the state variable $\left(\mathbf{x}_k^{(p)}\right)_{p,k}$. Given that the cost function $\ell$ steers the weights/biases to minimize the mismatches, and consequently the adjoints $\left(\mathbf{b}_k^{(p)}\right)_{p,k}$ then the admissible regularization is contributing to the gradient's norm minimization (Equ. 10), which is a desirable features for generalization purposes (Barrett & Dherin, 2020; Smith et al., 2021).

**The "granular" solution:** The admissible regularisation could be removed and this would lead to a "granular" solution which updates only the bias of the output layer $\boldsymbol{\eta}_n$. The granular solution is not practical and it is of no use for the neural network training.

**Hypotheses relaxation:** Some of the hypotheses of the ESN theorem could be relaxed: notably the invertibility of the output curvature: $\left\{\frac{\partial^2 \ell_R}{\partial \mathbf{x}_n^{(p)} \partial \mathbf{x}_n^{(p)}}\right\}$. Moore-Penrose inverse could be used instead. The output curvature is often positive definite and its inversion is simple for e.g. it is the identity for an $l^2$ cost function.

## 3 ADJUSTMENT OF THE EXACT STOCHASTIC NEWTON (ESN)

The ESN solution is exact and simple to compute; however, it diverges if implemented as it is. In this section, we first address the non-convexity issue and adjust the solution to become a descent direction. We also explore the stochasticity effect stemming from the samples randomness during the training process and provide an enhancement to the ESN. For the remainder of this paper we assume also that regularization is a of the same form as Equ. 4. We start this section by correcting the concave directions of the ESN.

### 3.1 ESCAPING THE SADDLES: A DESCENT DIRECTION

The minimal rank solution (Equ. 17) has some similarities with the gradient expression (Equ. 10), yet it is not a descent direction. To further illustrate this, we consider the inner product of the Newton and the gradient expressions layer by layer. For the weights, the product is always positive:

$$\langle \boldsymbol{N}_k, \boldsymbol{G}_k \rangle = \left\langle \mathbf{x}_{k-1}^{(p)}, \boldsymbol{\Lambda}_{k-1}^{(p)} \mathbf{x}_{k-1}^{(p)} \right\rangle = \left\| \mathbf{x}_{k-1}^{(p)} \right\|_{\boldsymbol{\Lambda}_{k-1}^{(p)}}^2 \geq 0. \tag{19}$$

However, most of the Newton biases corresponds to ascent directions :

$$\langle \boldsymbol{\eta}_k, \mathbf{g}_k \rangle = -\left\| \mathbf{x}_{k-1}^{(p)} \right\|_{\boldsymbol{\Lambda}_{k-1}^{(p)}}^2 + \mathbf{1}_{k=n} \left\| \frac{\partial \ell_R}{\partial \mathbf{x}_n^{(p)}} \right\|_{\left( \frac{\partial^2 \ell_R}{\partial \mathbf{x}_n^{(p)} \partial \mathbf{x}_n^{(p)}} \right)^{-1}}^2. \tag{20}$$

The last layer bias nature depends on the values of the regularization and the error. One of the main advantages of the Sifrian formulation is that we can treat the weights and biases separately and we can exactly pinpoint to the origin of non-convexity: the bias update. We choose to build our saddle-free variation by just changing the signs of the ascent direction. This manipulation is straightforward for all layers except the output bias update. The problematic term corresponds to the "granular" solution (i.e., no regularization), which overfits every sample. We chose to keep the "granular" solution as is and switch the sign of the remaining term. This guarantees that the update is a descent direction[5]. The saddle-free ESN is the following:

$$\begin{cases} \boldsymbol{N}_k = -\frac{\nabla F\left(\mathbf{a}_k^{(p)}\right) \mathbf{b}_k^{(p)}}{\left\| \nabla F\left(\mathbf{a}_k^{(p)}\right) \mathbf{b}_k^{(p)} \right\|_2^2} \left( \frac{\partial \ell_R}{\partial \mathbf{x}_{k-1}^{(p)}} \right)^T, \\[4mm] \boldsymbol{\eta}_k = \mathbf{1}_{k=n} \left[ \nabla F\left(\mathbf{a}_n^{(p)}\right) \right]^{-1} \left\{ \frac{\partial^2 \ell_R}{\partial \mathbf{x}_n^{(p)} \partial \mathbf{x}_n^{(p)}} \right\}^{-1} \frac{\partial \ell_R}{\partial \mathbf{x}_n^{(p)}} + \boldsymbol{N}_k \mathbf{x}_{k-1}^{(p)}. \end{cases} \tag{21}$$

The above convexity correction is significantly simpler than the currently existing methods such as the Gauss Newton truncation (e.g. Schraudolph, 2002; Martens, 2010) or the absolute Hessian (Dauphin et al., 2014). The convexity correction is the first step in improving the ESN. Another intrinsic issue in the above formulation is that we try to optimize a given sample without taking into consideration the yet "unknown" coming samples. The essence of stochasticity in still missing. In the next section, we include such an aspect into the ESN method.

### 3.2 RANDOMIZATION

The ESN solution we have proposed so far is exact but remains deterministic since it focuses on training a single pre-selected sample $p$. An intrinsic source of stochasticity in online learning stems from the randomness of the streamed samples. As such, it is preferable to use random variables to describe the dynamics of ESN. The two fundamental equations of ESN after discarding the "granular" solution, adding the white layer are:

$$\begin{cases} \left( \frac{\partial \ell_R}{\partial \mathbf{x}_{k-1}^{(p)}} \right)^T + \mathbf{y}_k^{(p)T} \boldsymbol{N}_k = 0, \quad \text{with} \quad \mathbf{y}_k^{(p)} = \nabla F\left(\mathbf{a}_k^{(p)}\right) \mathbf{b}_k^{(p)}, \\[4mm] \boldsymbol{N}_k \mathbf{x}_{k-1}^{(p)} + \boldsymbol{\eta}_k = 0. \end{cases} \tag{22}$$

---

[5]This point could be further justified with a first order Taylor development of the loss function.

Let's consider $\Omega = \mathbb{D}$ as the set of possible outcomes of the training samples[6] and randomize the sample index $p \to$ p. Aside from $\{\boldsymbol{N}, \boldsymbol{\eta}\}$ all the vectors in the above system become random variables. We solve the randomized Equ. 22 using the following least-squares criteria:

$$\operatorname{argmin}_{\{\boldsymbol{N}, \boldsymbol{\eta}\}} \int_{\Omega=\mathbb{D}} d\Omega\left(\mathrm{p}\right) \left\| \left(\frac{\partial \ell_R}{\partial \mathbf{x}_{k-1}^{(\mathrm{p})}}\right)^T + \mathbf{y}_k^{(\mathrm{p})T} \boldsymbol{N}_k \right\|_2^2 + \left\| \boldsymbol{N}_k \mathbf{x}_{k-1}^{(\mathrm{p})} + \boldsymbol{\eta}_k \right\|_2^2. \tag{23}$$

We changed the ESN into a least-squares problem, which is desirable, given the sizeable well-established literature about optimal unbiased estimators for this class of problems (Aitken, 1936; Kariya & Kurata, 2004). The minimisation could be done for each layer separately, which reduces the inverse problem dimensionality. The search for critical points is detailed in the Appendix (A.4), and yields the following randomized ESN system which involves a Sylvester equation (for $\boldsymbol{N}_k$) :

$$\begin{cases} \mathbb{E}_{\mathrm{p}}\left[ \mathbf{y}_k^{(\mathrm{p})} \mathbf{y}_k^{(\mathrm{p})T} \right] \boldsymbol{N}_k + \boldsymbol{N}_k \operatorname{Var}\left(\mathbf{x}_{k-1}^{(\mathrm{p})}\right) = -\mathbb{E}_{\mathrm{p}}\left[ \mathbf{y}_k^{(\mathrm{p})} \left(\frac{\partial \ell_R}{\partial \mathbf{x}_{k-1}^{(\mathrm{p})}}\right)^T \right], \\ \boldsymbol{N}_k \mathbb{E}_{\mathrm{p}}\left[ \mathbf{x}_{k-1}^{(p)} \right] + \boldsymbol{\eta}_k = 0. \end{cases} \tag{24}$$

We notice that the above system could considerably simplify if the state variables $\mathbf{x}_{k-1}^{(p)}$ is normalized i.e. null mean and identity variance. This aspect has been extensively explored in Deep Learning through batch, weight, layer and group normalization (Ioffe & Szegedy, 2015; Salimans & Kingma, 2016; Ba et al., 2016b; Wu & He, 2018). Normalizing the state variable corresponds to the following transformation:

$$\mathbf{s}_k^{(\mathrm{p})} \leftarrow \left( \sqrt{\operatorname{Var}\left(\mathbf{x}_k^{(\mathrm{p})}\right)} \right)^{-1} \left( \mathbf{x}_k^{(\mathrm{p})} - \mathbb{E}_{\mathrm{p}}\left[ \mathbf{x}_k^{(\mathrm{p})} \right] \right). \tag{25}$$

The Sylvester equation for the $\boldsymbol{N}_k$ could be entirely circumvented, by using the proposed Variance-Invariance Theorem described hereafter.

**Variance-Invariance Theorem:** In addition to the hypotheses of the ESN Thereom (Section 2.2). If $\forall k \in [1..n] : \operatorname{Var}\left(\mathbf{x}_{k-1}^{(\mathrm{p})}\right) \succ 0$, then the randomized ESN is invariant to the variances $\operatorname{Var}\left(\mathbf{x}_{k-1}^{(\mathrm{p})}\right)$ which can be substituted with the identity. The randomized ESN system becomes:

$$\begin{cases} \boldsymbol{N}_k = -\mathbb{E}\left[ \boldsymbol{I} + \mathbf{y}_k^{(\mathrm{p})} \mathbf{y}_k^{(\mathrm{p})T} \right]^{-1} \mathbb{E}\left[ \mathbf{y}_k^{(\mathrm{p})} \left(\frac{\partial \ell_R}{\partial \mathbf{x}_{k-1}^{(\mathrm{p})}}\right)^T \right], \\ \\ \boldsymbol{\eta}_k = \boldsymbol{N}_k \mathbb{E}\left[ \mathbf{x}_{k-1}^{(\mathrm{p})} \right]. \end{cases} \tag{26}$$

**Proof:** The demonstration builds on the invariance of the Newton method to affine transformations and is detailed step by step in the Appendix (A.5) □

## 3.3 DISCUSSION

**Expectations and estimators:** We used the expectation $\mathbb{E}$ extensively in this paper to derive the theoretical ESN solution. However, estimators would replace expectation during implementation. The performance and the computational cost would be directly related to the estimators choice: rank one, moving average, or even batch estimators are all possible with different compromises.

**Notes on convergence:** Several convergence results for the Newton method exist (e.g Bottou & Le Cun, 2005; Byrd et al., 2011; Meng et al., 2020). The ESN convergence fits well within the general framework developed by Sunehag et al. (2009), under some mild and non-restrictive conditions.

**To batch or not to batch:** After randomization, ESN could be used in batch mode. However, there are no guarantees that such a solution solves the "batch" Sifrian equations. Therefore, we focus only on using spot or historical data.

---

[6]A $\sigma-$field is more appropriate, but we use only a random space $\Omega$ for the sake of simplicity.

## 4 RESULTS

We validate the ESN method for FNNs, using a classification task on MNIST dataset (Lecun et al., 1998), and also an autoencoder training with the CURVES dataset (Hinton & Salakhutdinov, 2006). Leaky Relu and He initializer (He et al., 2016; Glorot & Bengio, 2010) are used for all the training. Further implementation details are available in the Appendix (A.6). We report the results hereafter:

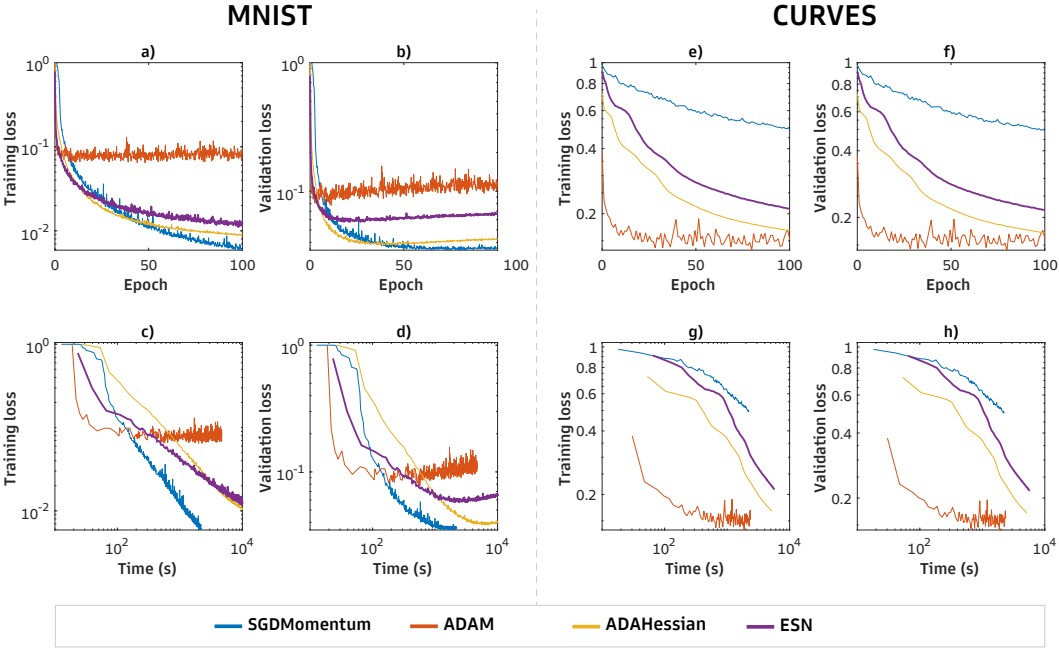

Figure 2: Training and validation loss as a function of both epochs and wall-clock time for classification (MNIST) and autoencoding (CURVES) with one seed. For MNIST classification, the architecture is $\{784, 256, 32, 10\}$. For CURVES, the autoencoder has the following architecture $\{784, 216, 64, 6, 64, 216, 784\}$. Constant learning rates are used 0.001 for ADAM (Kingma & Ba, 2014) and ADAHessian (Yao et al., 2020) and 0.01 for SGD and ESN.

## 5 CONCLUSION

This paper showed that it is possible to derive an exact expression of the stochastic second-order update for Deep Learning via a second-order Lagrangian which we called Sifrian, under some regularization constraints. We revisited backpropagation using Lagrange multipliers method and developed a new framework to better understand the mechanics of the Newton method at the stochastic level. This framework yielded unexpected insights on how to correct the non-convexity, the importance of the first layer, and how to get a closed-form solution. We further proceeded to randomize the solution, to better align the training with its stochastic nature and avoid overfitting. We tested the solution using simple training models; and despite the known limitations of the Newton method, the ESN (Exact Stochastic Newton) solution was able to perform as well as adaptive methods in a non-convex setting. ESN holds great potential but still requires further refinement for batch training, parameters selection and estimators choice. Generalization of this work to other type of networks is also a research venue which needs further investigation.

### REPRODUCIBILITY STATEMENT

The results of this paper are reproducible. Commented Matlab code is included in the supplementary material. All experiments are run on a workstation with eight cores 2.6 GHz CPU.

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

# A  APPENDIX

## A.1  $\mathcal{C}^2$ SIFRIAN EQUATIONS

The inner product used in the Sifrian is either the inner product of vectors or the trace product for matrices, which are equivalent after vectorization:

$$\langle \boldsymbol{A}, \mathbf{B} \rangle = \mathrm{trace}\left(\boldsymbol{B}^T \boldsymbol{A}\right) = \mathrm{vec}\left(\boldsymbol{B}\right)^T \mathrm{vec}\left(\boldsymbol{A}\right),$$

$$\langle \mathbf{x}, \mathbf{y} \rangle = \mathbf{x}^T \mathbf{y} = \mathrm{trace}\left(\mathbf{x}\mathbf{y}^T\right).$$

The derivation is carried through perturbation factorization for e.g.:

$$\mathcal{S}\left(\boldsymbol{W}_k + \delta \boldsymbol{W}_k\right) - \mathcal{S}\left(\boldsymbol{W}_k\right) \sim \left\langle \frac{\partial \mathcal{S}}{\partial \boldsymbol{W}_k}, \delta \boldsymbol{W}_k \right\rangle.$$

We carry the derivation and we report hereafter **the full Sifrian** equations for a $\mathcal{C}^2$ activation functions:

$$
\begin{cases}
(\mathbf{1}) \ -\sum_p \nabla F_k^{(p)} \left( \boldsymbol{\gamma}_k^{(p)} \mathbf{x}_{k-1}^{(p)T} + \mathbf{1}_{k=2..n} \mathbf{b}_k^{(p)} \boldsymbol{\zeta}_{k-1}^{(p)T} \right) \\[2mm]
+\sum_p \nabla^2 F_k^{(p)} \mathbf{b}_k^{(p)} \left[ \boldsymbol{N}_k \mathbf{x}_{k-1}^{(p)} + \boldsymbol{\eta}_k - \mathbf{1}_{k=2..n} \boldsymbol{W}_k \boldsymbol{\zeta}_{k-1}^{(p)} \right] \mathbf{x}_{k-1}^{(p)T} \\[2mm]
= \sum_p \nabla F_k^{(p)} \mathbf{b}_k^{(p)} \mathbf{x}_{k-1}^{(p)T}, \\[2mm]
(\mathbf{2}) \ \sum_p \nabla^2 F_k^{(p)} \mathbf{b}_k^{(p)} \left[ \boldsymbol{N}_k \mathbf{x}_{k-1}^{(p)} + \boldsymbol{\eta}_k - \mathbf{1}_{k=2..n} \boldsymbol{W}_k \boldsymbol{\zeta}_{k-1}^{(p)} \right] - \sum_p \nabla F_k^{(p)} \left( \boldsymbol{\gamma}_k^{(p)} + \mathbf{b}_k^{(p)} \right) = 0, \\[2mm]
(\mathbf{3}) \ \boldsymbol{\gamma}_k^{(p)} - \mathbf{1}_{k=1..n-1} \boldsymbol{W}_{k+1}^T \nabla F_{k+1}^{(p)} \boldsymbol{\gamma}_{k+1}^{(p)} + \sum_m \frac{\partial^2 \ell_R}{\partial \mathbf{x}_k^{(p)} \partial \mathbf{x}_m^{(p)}} \boldsymbol{\zeta}_m^{(p)}) \\[2mm]
+\mathbf{1}_{k=1..n-1} \boldsymbol{W}_{k+1}^T \nabla^2 F_{k+1}^{(p)} \mathbf{b}_{k+1}^{(p)} \left[ \boldsymbol{N}_{k+1} \mathbf{x}_k^{(p)} + \boldsymbol{\eta}_{k+1} - \mathbf{1}_{k=1..n-1} \boldsymbol{W}_{k+1} \boldsymbol{\zeta}_k^{(p)} \right] \\[2mm]
+\mathbf{1}_{k=1..n-1} \boldsymbol{N}_{k+1}^T \nabla F_{k+1}^{(p)} \mathbf{b}_{k+1}^{(p)} = 0, \\[2mm]
(\mathbf{4}) \ \boldsymbol{\zeta}_k^{(p)} - \mathbf{1}_{k=2..n} \nabla F_k^{(p)} \boldsymbol{W}_k \boldsymbol{\zeta}_{k-1}^{(p)} + \nabla F_k^{(p)} \left[ \boldsymbol{N}_k \mathbf{x}_{k-1}^{(p)} + \boldsymbol{\eta}_k \right] = 0.
\end{cases}
$$

## A.2  ESN RESOLUTION FOR $\mathcal{C}^2$ ACTIVATION FUNCTIONS

The stochastic case for $\mathcal{C}^2$ functions has the same type of simplifications. Comparing the first and second equations:

$$(\mathbf{1}) \ -\sum_p \nabla F_k^{(p)} \left( \boldsymbol{\gamma}_k^{(p)} \mathbf{x}_{k-1}^{(p)T} + \mathbf{1}_{k=2..n} \mathbf{b}_k^{(p)} \boldsymbol{\zeta}_{k-1}^{(p)T} \right)$$

$$+\sum_p \nabla^2 F_k^{(p)} \mathbf{b}_k^{(p)} \left[ \boldsymbol{N}_k \mathbf{x}_{k-1}^{(p)} + \boldsymbol{\eta}_k - \mathbf{1}_{k=2..n} \boldsymbol{W}_k \boldsymbol{\zeta}_{k-1}^{(p)} \right] \mathbf{x}_{k-1}^{(p)T}$$

$$= \sum_p \nabla F_k^{(p)} \mathbf{b}_k^{(p)} \mathbf{x}_{k-1}^{(p)T},$$

$$(\mathbf{2}) \ \sum_p \nabla^2 F_k^{(p)} \mathbf{b}_k^{(p)} \left[ \boldsymbol{N}_k \mathbf{x}_{k-1}^{(p)} + \boldsymbol{\eta}_k - \mathbf{1}_{k=2..n} \boldsymbol{W}_k \boldsymbol{\zeta}_{k-1}^{(p)} \right] - \sum_p \nabla F_k^{(p)} \left( \boldsymbol{\gamma}_k^{(p)} + \mathbf{b}_k^{(p)} \right) = 0,$$

This yields that $\mathbf{1}_{k=1..n-1}\boldsymbol{\zeta}_p(k) = 0$. The fourth equation state that $\left[\boldsymbol{N}_k \mathbf{x}_{k-1}^{(p)} + \boldsymbol{\eta}_k\right] = 0$ is always null except for the last layer. Let's focus on the last layer. Three equations allows to characterize it:

$$\begin{cases} (2)\ \mathbf{b}_n^{(p)} = \left(\nabla F_n^{(p)}\right)^{-1} \nabla^2 F_n^{(p)} \mathbf{b}_n^{(p)} \left[\boldsymbol{N}_n \mathbf{x}_{n-1}^{(p)} + \boldsymbol{\eta}_n\right] - \boldsymbol{\gamma}_n^{(p)}, \\[2mm] (3)\ \boldsymbol{\gamma}_n^{(p)} + \frac{\partial^2 \ell_R}{\partial \mathbf{x}_n^{(p)} \partial \mathbf{x}_n^{(p)}} \boldsymbol{\zeta}_n^{(p)} = 0, \\[2mm] (4)\ \boldsymbol{\zeta}_n^{(p)} + \nabla F_n^{(p)} \left[\boldsymbol{N}_n \mathbf{x}_{n-1}^{(p)} + \boldsymbol{\eta}_n\right] = 0. \end{cases}$$

If we assume that the $\mathcal{C}^2$ has a non null second derivative everywhere then, the previous three equations yield a specific value of $\left[\boldsymbol{N}_n \mathbf{x}_{n-1}^{(p)} + \boldsymbol{\eta}_n\right] = 0$ and also $\boldsymbol{\zeta}_n^{(p)}$:

Except the last layer we have: $\boldsymbol{\gamma}_k^{(p)} = -\mathbf{b}_k^{(p)}$, and a similar system to the piecewise affine case could be derived.

## A.3 EXAMPLE: A SIMPLE REGULARIZATION

In this section we give a concrete simple example to further explain the ESN method. In the case of an $l^2$ cost function and regularization defined by:

$$R(\mathbf{x}) = \frac{\lambda}{2} \sum_{p \in \mathbb{D}} \sum_{k=1..n-1} \left\langle \mathbf{x}_k^{(p)}, \mathbf{x}_k^{(p)} \right\rangle.$$

we have the following solution:

$$\begin{cases} \boldsymbol{N}_k = -\frac{\nabla F\left(\mathbf{a}_k^{(p)}\right) \mathbf{b}_k^{(p)} \mathbf{x}_{k-1}^{(p)T}}{\left\|\nabla F\left(\mathbf{a}_k^{(p)}\right) \mathbf{b}_k^{(p)}\right\|_2^2}, \\[4mm] \boldsymbol{\eta}_k = \mathbf{1}_{k=n} \left[\nabla F\left(\mathbf{a}_n^{(p)}\right)\right]^{-1} \left(\mathbf{d}^{(p)} - \mathbf{x}_n^{(p)}\right) - \boldsymbol{N}_k \mathbf{x}_{k-1}^{(p)}. \end{cases}$$

- The previous solution is straightforward to compute; the first order backpropagation provides all the elements.
- It is co-linear with the gradient except for the bias of the last layer.
- The adjoints $\mathbf{b}_k^{(p)}$) are impacted by choice of the regularization parameter $\lambda$.
- The minimal rank Newton solution is not a descent direction. Most of the biases correspond to an ascent direction.
- The squared norm division is problematic and will lead to divergent solution if the norm of the backpropagated errors is very small i.e. $\left\|\nabla F\left(\mathbf{a}_k^{(p)}\right) \mathbf{b}_k^{(p)}\right\|_2^2 \to 0$. A sufficient condition to avoid divergence consists in defining layer by layer regularization such as: $\lambda_{k-1} \propto \left\|\nabla F\left(\mathbf{a}_k^{(p)}\right) \mathbf{b}_k^{(p)}\right\|$.
- It is possible to remove the regularization i.e. $\lambda \to 0$, this yields a "granular" solution which states that the second order optimal solution consists in just adjusting the last layer bias with a slope corrected error: $\left[\nabla F\left(\mathbf{a}_n^{(p)}\right)\right]^{-1} \left(\mathbf{d}^{(p)} - \mathbf{x}_n^{(p)}\right)$. The granular solution is not suitable for neural network training, as it overfits every sample.

## A.4 PROOF: PROPOSITION RANDOMIZED ESN

We start from the randomized least-squares criteria.

$$\text{argmin}_{\{\boldsymbol{N},\boldsymbol{\eta}\}} \int_{\Omega=\mathbb{D}} d\Omega\,(\text{p}) \left\|\left(\frac{\partial \ell_R}{\partial \mathbf{x}_{k-1}^{(\text{p})}}\right)^T + \mathbf{y}_k^{(\text{p})T} \boldsymbol{N}_k\right\|_2^2 + \left\|\boldsymbol{N}_k \mathbf{x}_{k-1}^{(\text{p})} + \boldsymbol{\eta}_k\right\|_2^2.$$

The optimum is a critical point i.e. :

$$\forall k \; \left( \frac{\partial}{\partial \boldsymbol{N}_k}, \frac{\partial}{\partial \boldsymbol{\eta}_k} \right) = (0, 0) \,.$$

Each layer could be optimized separately which is desirable feature that reduce the dimensionality of the problem. We get the following derivatives:

$$\begin{cases} \frac{\partial}{\partial \boldsymbol{N}_k} \rightarrow \int d\Omega \; \left( \mathbf{y}_k^{(\mathrm{p})} \left( \frac{\partial \ell_R}{\partial \mathbf{x}_{k-1}^{(\mathrm{p})}} \right)^T + \mathbf{y}_k^{(\mathrm{p})} \mathbf{y}_k^{(\mathrm{p})T} \boldsymbol{N}_k + \boldsymbol{N}_k \mathbf{x}_{k-1}^{(\mathrm{p})} \mathbf{x}_{k-1}^{(\mathrm{p})T} + \boldsymbol{\eta}_k \mathbf{x}_{k-1}^{(\mathrm{p})T} \right) = 0, \\[2ex] \frac{\partial}{\partial \boldsymbol{\eta}_k} \rightarrow \int d\Omega \; \left( \boldsymbol{N}_k \mathbf{x}_{k-1}^{(\mathrm{p})} + \boldsymbol{\eta}_k \right) = 0 \end{cases}$$

Integration over $\Omega$ yields expectations and two sets of equations:

$$\begin{cases} \mathbb{E} \left[ \mathbf{y}_k^{(\mathrm{p})} \left( \frac{\partial \ell_R}{\partial \mathbf{x}_{k-1}^{(\mathrm{p})}} \right)^T \right] + \mathbb{E} \left[ \mathbf{y}_k^{(\mathrm{p})} \mathbf{y}_k^{(\mathrm{p})T} \right] \boldsymbol{N}_k + \boldsymbol{N}_k \mathbb{E} \left[ \mathbf{x}_{k-1}^{(\mathrm{p})} \mathbf{x}_{k-1}^{(\mathrm{p})T} \right] + \boldsymbol{\eta}_k \mathbb{E} \left[ \mathbf{x}_{k-1}^{(\mathrm{p})T} \right] = 0, \\[2ex] \boldsymbol{N}_k \mathbb{E} \left[ \mathbf{x}_{k-1}^{(\mathrm{p})} \right] + \boldsymbol{\eta}_k = 0. \end{cases}$$

We simplify the system by right multiplying the second equation with $\mathbb{E} \left[ \mathbf{x}_{k-1}^{(\mathrm{p})} \right]$ and substracting, we get:

$$\begin{cases} \mathbb{E} \left[ \mathbf{y}_k^{(\mathrm{p})} \mathbf{y}_k^{(\mathrm{p})T} \right] \boldsymbol{N}_k + \boldsymbol{N}_k \mathrm{Var} \left( \mathbf{x}_{k-1}^{(\mathrm{p})} \right) = - \mathbb{E} \left[ \mathbf{y}_k^{(\mathrm{p})} \left( \frac{\partial \ell_R}{\partial \mathbf{x}_{k-1}^{(\mathrm{p})}} \right)^T \right], \\[2ex] \boldsymbol{N}_k \mathbb{E} \left[ \mathbf{x}_{k-1}^{(\mathrm{p})} \right] + \boldsymbol{\eta}_k = 0. \end{cases}$$

The above expression corresponds to a Sylvester equation. The last two equations are the main result of the proposition.

### A.5 Invariance of ESN to normalisation:

In this section we demonstrate that the ESN is invariant to normalisation which would allow for a simpler resolution of the Sylvester Equation. Such a result is expected for a Newton method, we do the demonstration step by step. First let's start by reformulating the forward problem with the normalized state variable $\mathbf{s}$:

$$\mathbf{s}_k^{(\mathrm{p})} = \left( \sqrt{\mathrm{Var} \left( \mathbf{x}_k^{(\mathrm{p})} \right)} \right)^{-1} \left( \mathbf{x}_k^{(\mathrm{p})} - \mathbb{E}_{\mathrm{p}} \left[ \mathbf{x}_k^{(\mathrm{p})} \right] \right) = \boldsymbol{\Sigma}_k^{-1} \left( \mathbf{x}_k^{(\mathrm{p})} - \mathbf{m}_k \right) \,.$$

The original state variable is an affine transformation of the normalized one:

$$\mathbf{x}_k^{(p)} = \boldsymbol{\Sigma}_k \mathbf{s}_k^{(\mathrm{p})} + \mathbf{m}_k.$$

The matrix $\boldsymbol{\Sigma}_k$ is symmetric and positive. We assume further that it is definite i.e. invertible in this context. The forward model becomes:

$$\boldsymbol{\Sigma}_k \mathbf{s}_k^{(\mathrm{p})} + \mathbf{m}_k - F \left( \boldsymbol{W}_k \left( \boldsymbol{\Sigma}_{k-1} \mathbf{s}_{k-1}^{(\mathrm{p})} + \mathbf{m}_{k-1} \right) + \beta_k \right) \,.$$

The square-root of the variance and the expectation could be absorbed into the weights/biases:

$$\begin{cases} \boldsymbol{W}_k' = \boldsymbol{W}_k \boldsymbol{\Sigma}_{k-1}, \\[2ex] \boldsymbol{\beta}_k' = \boldsymbol{\beta}_k + \boldsymbol{W}_k \mathbf{m}_{k-1}. \end{cases}$$

With the above notations, the forward model becomes:

$$\boldsymbol{\Sigma}_k \mathbf{s}_k^{(\mathrm{p})} + \mathbf{m}_k = F \left( \boldsymbol{W}_k' \mathbf{s}_{k-1}^{(\mathrm{p})} + \boldsymbol{\beta}_k' \right) \,.$$

We use a new adjoint $\mathbf{u}$. The Lagrangian becomes:

$$\mathcal{L}(\mathbf{x}, \boldsymbol{W}, \beta, \mathbf{b}) = \ell_R + \sum_{p \in \mathbb{D}} \sum_{k=1..n} \left\langle \boldsymbol{\Sigma}_k \mathbf{s}_k^{(\mathrm{p})} + \mathbf{m}_k - F\left(\boldsymbol{W}_k' \mathbf{s}_{k-1}^{(\mathrm{p})} + \boldsymbol{\beta}_k'\right), \mathbf{u}_k^{(\mathrm{p})} \right\rangle.$$

The derivative w.r.t. $\mathbf{s}_k^{(\mathrm{p})}$ should be null, which yields the following backpropagation equation.

$$\frac{\partial \mathcal{L}}{\partial \mathbf{s}_k^{(\mathrm{p})}} = \boldsymbol{\Sigma}_k \mathbf{u}_k^{(p)} - \mathbf{1}_{k=1..n-1} \boldsymbol{W}_{k+1}'^{T} \nabla F\left(\mathbf{a}_{k+1}^{(p)}\right) \mathbf{u}_{k+1}^{(p)} + \boldsymbol{\Sigma}_k \frac{\partial \ell_R}{\partial \mathbf{x}_k^{(p)}} = 0.$$

Given the definition of $\boldsymbol{W}_{k+1}'$. The adjoint equation is the same as the original, hence $\boldsymbol{u} = \boldsymbol{b}$, and the first order adjoint is unmodified. The gradient is slightly modified:

$$\begin{cases} \boldsymbol{G}_k' = \frac{\partial \mathcal{L}}{\partial \boldsymbol{W}_k'} = -\sum_p \nabla F\left(\mathbf{a}_k^{(p)}\right) \mathbf{u}_k^{(p)} \mathbf{s}_{k-1}^{(\mathrm{p})T}, \\[2mm] \mathbf{g}_k' = \frac{\partial \mathcal{L}}{\partial \boldsymbol{\beta}_k'} = -\sum_p \nabla F\left(\mathbf{a}_k^{(p)}\right) \mathbf{u}_k^{(p)}. \end{cases}$$

Let's move to the Sifrian:

$$\mathcal{S}(\mathbf{s}, \mathbf{u}, \boldsymbol{W}', \boldsymbol{\beta}', \boldsymbol{G}', \mathbf{g}', \boldsymbol{\gamma}', \boldsymbol{\zeta}', \boldsymbol{N}', \boldsymbol{\eta}') = \sum_{k,p} \left\langle \boldsymbol{\Sigma}_k \mathbf{s}_k^{(\mathrm{p})} + \mathbf{m}_k - F\left(\boldsymbol{W}_k' \mathbf{s}_{k-1}^{(\mathrm{p})} + \boldsymbol{\beta}_k'\right), \boldsymbol{\gamma}_k'^{(p)} \right\rangle$$

$$+ \left\langle \boldsymbol{\Sigma}_k \mathbf{u}_k^{(p)} - \mathbf{1}_{k=1..n-1} \boldsymbol{W}_{k+1}'^{T} \nabla F\left(\mathbf{a}_{k+1}^{(p)}\right) \mathbf{u}_{k+1}^{(p)} + \boldsymbol{\Sigma}_k \frac{\partial \ell_R}{\partial \mathbf{x}_k^{(p)}}, \boldsymbol{\zeta}_k'^{(p)} \right\rangle$$

$$+ \sum_k \left\langle \boldsymbol{G}_k' + \sum_p \nabla F\left(\mathbf{a}_k^{(p)}\right) \mathbf{u}_k^{(p)} \mathbf{s}_{k-1}^{(\mathrm{p})T}, \boldsymbol{N}_k' \right\rangle$$

$$+ \left\langle \mathbf{g}_k' + \sum_p \nabla F\left(\mathbf{a}_k^{(p)}\right) \mathbf{u}_k^{(p)}, \boldsymbol{\eta}_k' \right\rangle.$$

The Sifrian equations are:

**(1)** $\sum_p \nabla F_k^{(p)} \left( \boldsymbol{\gamma}_k'^{(p)} \mathbf{s}_{k-1}^{(p)T} + \mathbf{u}_k^{(p)} \mathbf{s}_{k-1}^{(p)T} + \mathbf{1}_{k=2..n} \mathbf{u}_k^{(p)} \boldsymbol{\zeta}_{k-1}'^{(p)T} \right) = 0,$

**(2)** $\sum_p \nabla F_k^{(p)} \left( \boldsymbol{\gamma}_k'^{(p)} + \mathbf{u}_k^{(p)} \right) = 0,$

**(3)** $\boldsymbol{\Sigma}_k \boldsymbol{\gamma}_k'^{(p)} - \mathbf{1}_{k<n} \boldsymbol{W}_{k+1}'^{T} \nabla F_{k+1}^{(p)} \boldsymbol{\gamma}_{k+1}'^{(p)} + \sum_m \boldsymbol{\Sigma}_k \frac{\partial^2 \ell_R}{\partial \mathbf{x}_k^{(p)} \partial \mathbf{x}_m^{(p)}} \boldsymbol{\Sigma}_m \boldsymbol{\zeta}_m'^{(p)} + \mathbf{1}_{k<n} \boldsymbol{N}_{k+1}'^{T} \nabla F_{k+1}^{(p)} \mathbf{u}_{k+1}^{(p)} = 0,$

**(4)** $\boldsymbol{\Sigma}_k \boldsymbol{\zeta}_k'^{(p)} - \mathbf{1}_{k=2..n} \nabla F_k^{(p)} \boldsymbol{W}_k' \boldsymbol{\zeta}_{k-1}^{(p)} + \nabla F_k^{(p)} \left( \boldsymbol{N}_k' \mathbf{s}_{k-1}^{(p)} + \boldsymbol{\eta}_k' \right) = 0.$

We follow the same steps of the ESN Theorem demonstration, add the white layer and we remove the granular solution. We finally get the following stochastic system:

$$\begin{cases} \left(\frac{\partial \ell_R}{\partial \mathbf{x}_k^{(p)}}\right)^T \boldsymbol{\Sigma}_k + \left(\nabla F\left(\mathbf{a}_{k+1}^{(p)}\right) \mathbf{u}_{k+1}^{(p)}\right)^T \boldsymbol{N}_{k+1}' = 0, \\[2mm] \boldsymbol{N}_k' \mathbf{s}_{k-1}^{(p)} + \boldsymbol{\eta}_k' = 0. \end{cases}$$

The randomization yields,

$$\begin{cases} \mathbb{E}\left[\mathbf{y}_k^{(\mathrm{p})} \mathbf{y}_k^{(\mathrm{p})T}\right] \boldsymbol{N}_k' + \boldsymbol{N}_k' \mathrm{Var}\left(\mathbf{s}_{k-1}^{(\mathrm{p})}\right) = -\mathbb{E}\left[\mathbf{y}_k^{(\mathrm{p})} \left(\frac{\partial \ell_R}{\partial \mathbf{x}_{k-1}^{(\mathrm{p})}}\right)^T\right] \boldsymbol{\Sigma}_k, \\[2mm] \boldsymbol{N}_k' \mathbb{E}\left[\mathbf{s}_{k-1}^{(\mathrm{p})}\right] + \boldsymbol{\eta}_k' = 0 \end{cases}$$

The $s$ is normalized hence:

$$\begin{cases} \text{Var}\left(\mathbf{s}_{k-1}^{(\text{p})}\right) = \boldsymbol{I}, \\[2mm] \mathbb{E}\left[\mathbf{s}_{k-1}^{(\text{p})}\right] = 0. \end{cases}$$

The randomized, ESN with the normalized state variable has the following solution:

$$\begin{cases} \boldsymbol{N}_k' = -\mathbb{E}\left[\boldsymbol{I} + \mathbf{y}_k^{(\text{p})}\mathbf{y}_k^{(\text{p})T}\right]^{-1} \mathbb{E}\left[\mathbf{y}_k^{(\text{p})}\left(\frac{\partial \ell_R}{\partial \mathbf{x}_{k-1}^{(\text{p})}}\right)^T\right] \boldsymbol{\Sigma}_k, \\[3mm] \boldsymbol{\eta}_k' = 0 \end{cases}$$

We revert to the original variables from Equ. A.5, add convexity correction and we get the following solution:

$$\begin{cases} \boldsymbol{N}_k = -\mathbb{E}\left[\boldsymbol{I} + \mathbf{y}_k^{(\text{p})}\mathbf{y}_k^{(\text{p})T}\right]^{-1} \mathbb{E}\left[\mathbf{y}_k^{(\text{p})}\left(\frac{\partial \ell_R}{\partial \mathbf{x}_{k-1}^{(\text{p})}}\right)^T\right], \\[3mm] \boldsymbol{\eta}_k = \boldsymbol{N}_k\mathbb{E}\left[\mathbf{x}_{k-1}^{(\text{p})}\right] \end{cases}$$

### A.6 IMPLEMENTATION DETAILS:

In this section we specify technical details of our implementation. First, we remove the "granular" terms from our solution, since they do overfit each sample by construction.:

$$\begin{cases} \boldsymbol{N}_k = -\mathbb{E}\left[\boldsymbol{I} + \mathbf{y}_k^{(\text{p})}\mathbf{y}_k^{(\text{p})T}\right]^{-1} \mathbb{E}\left[\mathbf{y}_k^{(\text{p})}\left(\frac{\partial \ell_R}{\partial \mathbf{x}_{k-1}^{(\text{p})}}\right)^T\right], \\[3mm] \boldsymbol{\eta}_k = \boldsymbol{N}_k\mathbb{E}\left[\mathbf{x}_{k-1}^{(\text{p})}\right] \end{cases}$$

ESN is solved for each layer independently, which means that the inverted matrices are a scale smaller than the empirical Fisher matrix for e.g.. To further illustrate this point, let's consider, for e.g., the feedforward network with layers sizes $[784, 256, 32, 10]$ (MNIST classification): the empirical Fisher matrix would have a size of 210298x210298, while the biggest matrix inversion for ESN would be only 256x256. In practice, the computation requires initially two expectations $\left(\mathbb{E}\left[\mathbf{y}_k^{(\text{p})}\left(\frac{\partial \ell_R}{\partial \mathbf{x}_{k-1}^{(\text{p})}}\right)^T\right], \mathbb{E}\left[\mathbf{x}_{k-1}^{(\text{p})}\right]\right)$ which could be estimated using exponentially decaying moving averages (Similar to ADAM). The product with the non-centered precision matrix $\mathbb{E}\left[I + \mathbf{y}_k^{(\text{p})}\mathbf{y}_k^{(\text{p})T}\right]^{-1}$, could be approximated in several ways. We list hereafter at least three methods:

- Direct method: the non-centered covariance matrix could be estimated using an exponentially decaying moving average and inverted directly given the small sizes.

- Recursive Least-Squares (RLS): A sliding window could be used to update recursively the estimation of $N(k)$ using Sherman-Morrison formula with an initialization of the precision matrix at $\boldsymbol{I}$. Such a method is a classic of online learning and is similar to Kalman filtering.

- Rank one method: instead of involving the historical data, the simplest/crudest estimation of the covariance matrix is based on the spot/instantaneous information. The precision matrix could be derived using Sherman-Morrison formula (applied once):

$$\left(\boldsymbol{I} + \mathbf{y}_k^{(\text{p})}\mathbf{y}_k^{(\text{p})T}\right)^{-1} = \left(\boldsymbol{I} - \frac{\mathbf{y}_k^{(\text{p})}\mathbf{y}_k^{(\text{p})T}}{1 + \left\|\mathbf{y}_k^{(\text{p})}\right\|_2^2}\right).$$

**Probability distribution/weights:** It is common to use weights in the context of least-squares regression. For Deep Learning, we could increase the importance of samples that are not well assimilated into the network, i.e., high errors(or vice-versa). This is possible for e.g. by choosing probabilities/weights proportional to errors' norms: $w^{(p)} \sim \left\| \mathbf{x}_n^{(p)} - \mathbf{d}^{(p)} \right\|_2^2$. The ESN regression equations become:

$$
\begin{cases}
\boldsymbol{N}_k = -\mathbb{E}\left[ w^{(p)}\boldsymbol{I} + w^{(p)}\mathbf{y}_k^{(\mathrm{p})}\mathbf{y}_k^{(\mathrm{p})T} \right]^{-1} \mathbb{E}\left[ w^{(p)}\mathbf{y}_k^{(\mathrm{p})} \left( \frac{\partial \ell_R}{\partial \mathbf{x}_{k-1}^{(\mathrm{P})}} \right)^T \right], \\
\\
\boldsymbol{\eta}_k = \boldsymbol{N}_k \frac{\mathbb{E}\left[ w^{(p)}\mathbf{x}_{k-1}^{(\mathrm{p})} \right]}{\mathbb{E}\left[ w^{(p)} \right]}
\end{cases}
$$

**Effects of the admissible regularization:** We stated in this paper that the admissible regularisation $R(\mathbf{x})$ reduces the norm of the state variable $\mathbf{x}$. To further illustrate this statistical effect, we will use MNIST classification as an example, with regularized and un-regularized (normal) stochastic gradient descent method. For the sake of simplicity we consider the following form of regularisation:

$$
R\left( \mathbf{x} \right) = \frac{1}{2} \sum_{p \in \mathbb{D}} \sum_{k=1..n-1} \lambda \left\langle \mathbf{x}_k^{(p)}, \mathbf{x}_k^{(p)} \right\rangle .
$$

The architecture of the FNN is the same from the experimental section i.e. $\{784, 256, 32, 10\}$. We select a learning rate of $\mu = 0.001$ and various values of $\lambda = \{0, 10^{-2}, 10^{-3}, 10^{-4}, 10^{-5}\}$.

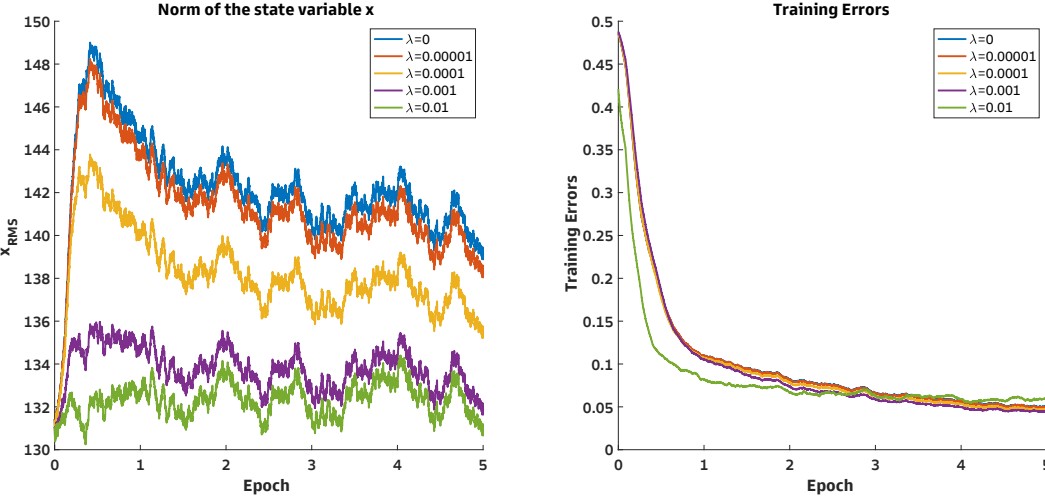

Figure 3: Effect of the admissible regularization on the norm of the state variable $\sum \left\| \mathbf{x}_k^{(p)} \right\|_2^2$ for different values of $\lambda = \{0, 10^{-2}, 10^{-3}, 10^{-4}, 10^{-5}\}$. SGD is used as the training method. The higher values of $\lambda$ reduce further the norm of the state variable $\mathbf{x}$. Training cost is reported in the right side of the figure.

