# OpenReview forum: "Exact Stochastic Newton Method for Deep Learning: the feedforward networks case."
_ICLR.cc/2022/Conference — ICLR 2022 Submitted_

### Official Review · Reviewer_CD8M · 2021-10-27

**Correctness:** 2
**Technical Novelty And Significance:** 1
**Empirical Novelty And Significance:** 2
**Recommendation:** 3
**Confidence:** 4

**Main Review:**

This manuscript is among the many proposals of applying a second-order method to train neural network models. The problem being tackled is for sure of great interest, but unfortunately, this manuscript fails to provide a meaningful solution to this open problem.
The major issues of the manuscript include:

1. Both the numerical result and the theoretical contributions are weak. Usually we expect a paper with either a strong theoretical result or outstanding numerical performance. Unfortunately, the numerical results show that the proposed method performs significantly inferior to existing methods, even in terms of the number of epochs and in terms of the training objective, for which usually second-order methods stand out. From the theoretical side, the auxiliary function for deriving the algorithm is not well-motivated, and the derived algorithm, although directly associated with the auxiliary function, has only weak connection with the original objective to minimize.

2. Another issue is that there is no convergence guaantees for the proposed algorithm. The authors claimed that the framework of Sunehag et al. (2009) could be applied, but the assumptions in that work require a unique global minimizer, which is not verifiable at all for neural network models.

3. I also have strong concerns against the claim of escaping saddles in Section 3.1. Usually in nonconvex optimization like the training of neural network models, escaping saddle points is referred to preventing the iterates from trapped in saddle points of the training objective, but the discussion in sec 3.1 is simply discussing how to obtain a descent direction in general. The terminology is quite confusing.

4. Overall speaking, the motivation of the proposed auxiliary function and of the algorithm is weak and the readability of the paper can be significantly improved. It was extremely hard for me to understand what the authors would like to express, but the concepts could actually be made much more intuitive. Even in the preliminary part of introducing backpropagation, the path the authors took is abstruse, while back propagation itself should be a simple concept by viewing from the simple chain rules. The proposed auxiliary function is also hard to understand the intuition behind and its connection with the optimization of the original objective. What the auxiliary function describes and why we want to solve for the point satisfying (12) are not clearly described nor motivated, and clearly its connection to finding a second-order step for the training objective is weak. The modification of the update step to make it a descent one also looks like quite arbitrary, and if the original update direction is "Newton", this change for sure destroys the meaning of the derivation.

I have read the authors' responses, and some misunderstandings of the regularizer are clarified. But the quality of the work still remains a huge concern and my reccomendation for this manuscript remains unchanged.



**Summary Of The Paper:**

This manuscript proposes to apply a Newton-type algorithm for training feed-forward neural network models.
When the regularizer added to the optimization problem satisfies certain additional assumptions, the algorithm can be implemented efficiently with additional assumptions on the activation function, including monotonicity and piecewise affinity.


**Summary Of The Review:**

The manuscript is poorly motivated and not clearly written, and the proposed algorithm has  neither strong numerical nor sound theoretical support.

---

> ### Author Response · Authors · 2021-11-10
> **Misunderstanding**
>
> We thank the reviewer for the comments and feedback. We would do our best to elucidate any misunderstandings.
>
> The reviewer described our method as weak and the algorithm as weak.
>
> First, we would like to emphasize the paper's novelty:  **we efficiently leveraged the Hessian matrix information without any approximation/truncations or divergence issues for a large-scale inversion problem. We provided full proof and demonstrated that the new concepts are properly functional.** We acknowledge that the experimental part needs further polishing, which should be expected given the novelty of the method (Sifrian). We believe that this paper holds a significant contribution for second-order optimization for large scale inversion and is nothing short of a breakthrough.
>
> We believe there is some deep misunderstanding:
>
> Please let’s start by stating that **the Hessian is not block-diagonal.** There is no such statement anywhere in the paper, especially since we did not formulate the Hessian explicitly. The only block diagonal matrix that appears in our paper is $\Lambda$ (Equation 3) and it is not the Hessian. The Hessian is what is used to rescale the gradient and not $\Lambda$ from equation (3).
> For further clarification: the Hessian H appears in Equation (15).
>
>
> >1.	Both the numerical result and the theoretical contributions are weak. (...) Unfortunately, the numerical results show that the proposed method performs significantly inferior to existing methods, even in terms of the number of epochs and in terms of the training objective, for which usually second-order methods stand out. Clearly, the results show that the proposed method even failed to converge in some simple tasks, further showing that the proposed method is problematic. From the theoretical side, the auxiliary function for deriving the algorithm is not well-motivated, and the derived algorithm, although directly associated with the auxiliary function, has only weak connection with the original objective to minimize.
>
> * “Significantly inferior”: We are using logarithmic scales, and the difference are very small in absolute value.
> * “failed to converge” We do not have any divergence in the figures… We are performing better than ADAM for MNIST…
>
> >2.	The requirement for the regularizer to make the Hessian of the whole training objective (instead of just the regularizer) cross-talk free is extremely stringent, and barely practical. With such a condition imposed on the objective, clearly the application of Newton's method can be done with very low computational cost, but this requirement is almost meaningless for any possible application. The remaining simplification of the Newton direction calculation also relies on the special structure of the activation function, which is quite straightforward in general.
>
> We believe there is a major misunderstanding: the Hessian is not crosstalk-free. We are not rescaling the gradient with the inverse of $\Lambda$. Any activation function could be used if its gradient, which is a second-order tensor (matrix), exists and is invertible.
>
> >3.	I also have strong concern against the claim of escaping saddles in Section 3.1. Usually in nonconvex optimization like the training of neural network models, escaping saddle points is referred to preventing the iterates from trapped in saddle points of the training objective, but the discussion in sec 3.1 is simply discussing how to obtain a descent direction in general. The terminology is quite confusing.
>
> Newton method is attracted to saddle-points and behaves around them as if they were minima. This is a major flaw of the Newton method. Descent directions are not attracted to saddles…
>
> > ...the path the authors took is abstruse, while back propagation itself should be a simple concept by viewing from the simple chain rules. (...). What the auxiliary function describes and why we want to solve for the point satisfying (12) are not clearly described nor motivated, and clearly its connection to finding a second-order step for the training objective is weak...
>
> We apologize but there is a critical misunderstanding… We are not solving Equation (12). (12) describes the equilibrium of the Sifrian from which we derive equations (16). Solving (16) gives you the Newton direction, but it is challenging. So, we solve only the stochastic case (one sample at a time). The solution needs further convexity correction, and must consider the randomness of the streamed samples… We are performing better than ADAM for MNIST and better than SGD+ Momentum in CURVES. The scales are logarithmic and the difference in performance is small across all methods. The Lagrangian multipliers is not an abstruse method in our opinion: it is much simpler than the chain rule approach and it is more suitable for generalization to second-order.
>
>  We cannot help but feel there is a major misunderstanding, we will be glad to provide more details and information to dispel any ambiguity.

---

### Official Review · Reviewer_wxhq · 2021-11-02

**Correctness:** 2
**Technical Novelty And Significance:** 2
**Empirical Novelty And Significance:** 2
**Recommendation:** 3
**Confidence:** 3

**Main Review:**

Overall assessment
==================

Overall, I find the paper hard to read and follow, in particular due to the notations (see section bellow). The introduction of the Sifrian is not really justified (**Q1**). It seems to derive from a splitted problem with auxillary variables but this is never explicited. In particular, I believe there is a link with ADMM training that use such auxillary variable which should be explicited (**Q2**). Moreover, the results only apply to very specific networks and these constraints are not discussed (**Q3**) and the numerical experiments are not really convincing (**Q4**). For these reasons, I feel that the paper is not ready for publication and I recomend rejection.

**Q1:** Overall, I feel that the approach is not well justified in the current manuscript. In the Sifrian, it is unclear why the loss function is omitted? It is also unclear why one can choose in Eq.(14) quantities to be equals to what is proposed. In Eq.(15), it is not clear why $H$ would be the Hessian. Indeed, if $G,g$ were the gradient, it would be the Hessian but here, $G$ and $g$ are variables so it is unclear why it is exactly the Hessian. Overall, I feel that these missing justification is mainly due to an unclear exposition but this makes it very hard to get an intuition on why this method should work.

**Q2:** The proposed approach looks very similar to ADMM training for neural network (see for instance [A]). This paper should be mentionned and the main difference with the proposed method shoudl be highlighted.  Also, could you explain what is the main advantage of the Sifrian compared to the Augmented Lagrangian?

**Q3:** The assumptions for this method are not standard and are scattered across the paper. This makes it hard to know when this method can be applied. Indeed, the specific regularisation, the need for a white layer, a strictly monotonous do not appear in a common place when exposing the contribution. Moreover, the limitation they introduce are not really discuss.

**Q4:** The numerical evalution of the method is not really convincing. The scale of the considered problem is really small (MNIST and CURVE). The proposed method is not performing better than the other methods, neither in iteration or in time. For the other methods, the choice of parameters is not really discussed (for instance, is the `SGDMomentum` method slow on CURVE beccause of the choice of step size). Also, the implementation of the proposed method also include steps that are not discussed in the main manuscript to avoid overfitting, using moving average with an extra parameter. While this probably helps with variance reduction, discussing this in the main part of the paper seems necessary.


Minor comments, nitpicks and typos
----------------------------------

- **M1:** The need for the white layer seems very artificial. I would be surprised that it cannot be removed by specifying this constraint in the Sifrian.

- p.2: note$^1$ -> a CNN is a FNN. It would be better to say that while only dense networks are considered, this also applies to CNN.
- p.2 _notation $x_p(n)$ -> usually, using exponent notation $x^{(p)}$ for sample number makes it more readable, to avoid collapsing with coordinate selection and layer numbering in the compact notation.
- p.2 _"$p$ designs one single pattern"_ -> $p$ designs a single sample.
- p.2 _Compact notation_: The 2 sets of notations make it confusing and I think only using the compact notation would make the paper clearer.
- Definition.1: use a definition environment to make it clearer where the definition starts and stops.
- Eq.(4): it is not clear immediatly that $x$ is the activation of all layers.
- p.4 _"Our approach to extend the Lagrangian consists of"_ -> "consits in".
- Definition.2: I dont understand why you did not use $b_p$ instead of $\gamma_p$ to make the notation more consistent with Eq.(5)? I think this is the same term as in the Lagrangian and not changing this would help the reader follow.
- Eq.(12): I think there is a missing $g(W, \beta, x, b)$.
- p.5: _The first equation becomes_ -> Not sure why $b_{p, k}$ disappears and why the indexes  are not `2..n`.
- p.7: _"which by construction overfits to every pattern"_ -> overfits every sample.
- p.7: _"The last layer bias nature depend"_ -> depends.
- Eq.(25): put variable relatively to which you minimize under the argmin.


Extra References
----------------

[A] Taylor, G., Burmeister, R., Xu, Z., Singh, B., Patel, A. and Goldstein, T., 2016. [_Training neural networks without gradients: A scalable admm approach._](https://arxiv.org/pdf/1605.02026.pdf) In ICML (pp. 2722-2731).


**Summary Of The Paper:**

This paper proposes a stochastic second order method to train neural network under some specific regularisation criterion. The method is based on the Sifrian, an extension of the Lagrangian that splits the definition of the gradient of each layer's parameter as different constraints with their own multiplier. Solving the best direction from the Sifrian is complicated in the general case but can be done when considering only one sample. This allows for a stochastic algorithm to train the neural network. Finally, some limited  numerical experiments are conducted.

**Summary Of The Review:**

Overall, I find the paper hard to read and follow, in particular due to the notations. The introduction of the Sifrian is not really justified (**Q1**). It seems to derive from a splitted problem with auxillary variables but this is never explicited. In particular, I believe there is a link with ADMM training that use such auxillary variable which should be explicited (**Q2**). Moreover, the results only apply to very specific networks and these constraints are not discussed (**Q3**) and the numerical experiments are not really convincing (**Q4**). For these reasons, I feel that the paper is not ready for publication and I recomend rejection.

---

> ### Author Response · Authors · 2021-11-10
> **SIFRIAN, ADMM and Limitations**
>
> First, we would like to thank the reviewer for the comments and feedback.
> We will be glad to consider the suggested corrections and notations change.
>
> We also would like to first highlight that the reviewer believes that our presented work and particularly the Sifrian concept are linked to the augmented Lagrangian and ADMM.  In our opinion, such a link is rather fragile. ADMM is a well-established method that alternates the optimization of primal and dual variables. Separability is an important concept in ADMM, and some properties of our regularization could be considered as a separability condition (crosstalk free). Nevertheless, with the Sifrian we directly solve the stochastic Newton direction. We reiterate again that the link with ADMM is non-existent.
>
> **This paper holds a significant contribution for second-order optimization:  we efficiently leveraged the Hessian matrix information without any approximation/truncations or divergence issues for Deep Learning (FNN) in the stochastic case. We provided full proof and demonstrated that the new concepts are properly functional. We acknowledge that the experimental part needs further polishing, which should be expected given the novelty of the method (Sifrian).**
>
> We answer the issues raised by the reviewer point by point hereafter.
> >Q1
>
> Omitting the cost function is fundamental to defining the Sifrian.  We specifically avoided augmented Lagrangian for the second order. Eq (14) is about choice, and this whole paper is about choice. In a sense, we are replicating the same concepts of Lagrange multipliers demonstration with different variables. For Eq (15), H is the definition of the Hessian. From Eq (12) onwards, G and g (later comment: missing but I will add it) have special values and are functions of the weights, x, and the adjoint b…  The intuition is that the Sifrian can be used to reconstruct the equation Hessian x Newton=Gradient.
> We hope that you would appreciate the above intricacies and we are willing to provide further details if needed.
> >Q2
>
> The Sifrian has no saddle, it is null, always null and its derivatives are null when the forward equations, backward and the gradient definition are verified.  The Sifrian is different from the Lagrangian. We understand that the appellation “second-order Lagrangian” is misleading, but we had no other concept to compare to.
> Thank you for the reference but it is not directly related at this stage.
> The method is applicable to FNN and CNN, which is a reasonable start for a novel method, and we have been upfront about the applicability of our method from the beginning.
> >Q3
>
>  We acknowledge that the paper is written sequentially, and each section builds upon its predecessors. This is due to the fact that Newton method issues appear sequentially at each stage, and we are solving them as they appear.
> * White layer: we would like to highlight that such a problem is far from trivial and raises the question of how it is possible for Hessian-Free methods (HF) to properly function in the first place. Our answer is that HF does not work unless damping is injected. This is an important result of our paper. The white layer does not change the networks and allows updating the first layer.
> * Limitations, the piece affine condition could be relaxed and there is a solution for C2  functions in the appendix. Strict monotonicity is necessary to invert ($\nabla F$) and it is not restrictive in our opinion (a small linear term could be added).
> >Q4
>
> Our primary goal for the numerical evaluation is to provide a stochastic Newton method that uses the Hessian information without truncations and that does not diverge.  We understand that our results require further polishing:  we aimed only to validate the theoretical solution with well-established datasets such as MNIST or CURVES. The implementation details are fully included in the appendix with a reproducible code in the supplementary material.
>
> > Minor comments, nitpicks and typos
>
> Thank you for the minor comments, we corrected most of them.
> >*	M1: The need for the white layer seems very artificial. I would be surprised that it cannot be removed by specifying this constraint in the Sifrian.
>
> We do not share the reviewer's opinion: the white layer does not change the network and it fills the structural gap. Otherwise, there is no rule on how to update the first layer. Besides, we do not think that adding constraints on the Sifrian would be helpful... We would be glad to further discuss this point…
>
> >*	Definition.2: I dont understand why you did not use bp instead of γp to make the notation more consistent with Eq.(5)? I think this is the same term as in the Lagrangian and not changing this would help the reader follow.
>
> $b_p$ is the adjoint of 1st order and $\gamma_p$ is the adjoint of second order, there is no apriori reason why they would be the same. Later we find that $b_p=-\gamma_p$. This equality can not be predicted or used from the start.

---

### Official Review · Reviewer_Qnsb · 2021-11-02

**Correctness:** 3
**Technical Novelty And Significance:** 1
**Empirical Novelty And Significance:** 1
**Recommendation:** 3
**Confidence:** 4

**Main Review:**

In summary, my opinion is that the paper falls well short of its claims, for the following reasons:

1) The approximations involved are sometimes crude and often not well motivated. For example:
- What the authors call "stochastic case", is truly the "batch size = 1" case. I do not expect training with one data point per update to work in reasonable circumstances.
- I think approximating the inverse of H by knowing just its product with a single vector is unreasonable. Basically, we know the curvature of the loss along one direction only (which we can't even control)! I don't think there is any point in doing that, one is better served by using standard diagonal approximations of the Hessian or the Fisher matrix, as done in several previous work. In that case we actually know the curvature along several directions.
- The fact that only the bias, and not the weights, contribute to divergence of the Netwon's update worries me a lot, I don't expect that to happen in any reasonable circumstance. It suggests that all the approximations made are highly unrealistic.
- Part of the motivation is to go beyond the approximations made in previous work, but the authors end up making them anyway towards the end, e.g. the K-FAC like approximation.
- The assumption that the effect of variance is negligible in Eq.(27) is completely unjustified.


2) The writing of the paper is quite poor, and it goes worse and worse towards the end, with several typos and confusing statements. It looked like the authors struggled to finish the paper in time for the submission deadline. For example:
- I appreciate that the authors provide Section 1 to remind readers how to derive backpropagation using lagrange multipliers. That is very intuitive, introducing auxiluary variables and lagrange multipliers does not change the final result. However, when moving to section 2, it is completely unclear where the Newton's update comes from.
- Several sections towards the end of the paper are cunfusing and out of context, such as: "Probability distribution/weights", "Notes on convergence", "To batch or not to batch".
- Notation and nomenclature are highly confusing. Besides using symbols that are different from standard convention, which is maybe OK, they use confusing wording: "positive symmetric matrices" instead of "positive definite matrices", "adjoint state vectors" instead of "lagrange multipliers".

3) Finally, the "Results" section: by just looking at the plots, it seems that the proposed method is worse than most of competitors. Even worse, there is no comment or interpretation to make sense of the results.



**Summary Of The Paper:**

This paper claims that it is possible to compute the Newton method's update exactly for deep neural networks (multi-layer perceptrons).
The motivation is that Newton's method, as a second-order optimizer that includes loss curvature information, should improve upon first-order optimizers, such as gradient descent, that use only first derivatives (gradient) of the loss. Second order methods tipically converge in a smaller number of iterations, but each update is expensive to compute and that's why they are not widely used in practice.
It must be noted that Newton's method is almost never mentioned in the context of Neural Networks (NN), because it is only guaranteed to converge for convex loss functions (which is clearly not the case for NN). However, the authors claim they have some tricks to fix that.

**Summary Of The Review:**

Given the above points 1, 2, and 3 I recommend rejection

---

> ### Author Response · Authors · 2021-11-10
> **Comments**
>
> The authors thank the reviewer for the feedback.
> Nevertheless, some of the reviewer’s comments are not precise and we hope we could elucidate any misunderstanding.
>
> First, we would like to emphasize the paper's novelty:  **we efficiently leveraged the Hessian matrix information without any approximation/truncations or divergence issues for a large-scale inversion problem. We provided full proof and demonstrated that the new concepts are properly functional.** We acknowledge that the experimental part needs further polishing, which should be expected given the novelty of the method (Sifrian). We believe that this paper holds a significant contribution to second-order optimization for large-scale inversion and is nothing short of a breakthrough.
>
> We answer the raised issues hereafter.
>
> The reviewer is mostly describing the Newton method in a deterministic context...
> >It must be noted that Newton's method is almost never mentioned in the context of Neural Networks (NN) because it is only guaranteed to converge for convex loss functions (which is clearly not the case for NN). However, the authors claim they have some tricks to fix that.
>
> Such a claim is arguable and we think it is not accurate because there is extensive published research on second-order methods for training Neural Networks during the last decade. The first page contains numerous references to papers that use the Newton-type method and variations for NN training.
>
> >*	I do not expect training with one data point per update to work in reasonable circumstances.
>
> We respectfully disagree and we would refer for e.g., to the early work of Bottou or to the paper below (and references therein).
>
> https://leon.bottou.org/slides/onepass/onepass.pdf
>
> (Nakkiran et al, 2020). The Deep Bootstrap Framework:  https://openreview.net/pdf?id=guetrIHLFGI
>
> In this paper, we characterized the Newton direction in general and solved it exactly in the stochastic case. Diagonal approximations are not novel and are just approximations. We backed our results with complete proofs, and we would reconsider our method if there were any issues with the demonstrations.
>
> >*	The fact that only the bias, and not the weights, contribute to divergence of the Netwon's update worries me a lot, I don't expect that to happen in any reasonable circumstance. It suggests that all the approximations made are highly unrealistic.
>
> We understand the unexpected/surprising nature of our results, and how the “complicated” problem of convexity correction is handled with a simple sign change. This is an important result of this paper which we fully demonstrated. Please let us know if there are any mistakes in the demonstration.
>
> >*	Part of the motivation is to go beyond the approximations made in previous work, but the authors end up making them anyway towards the end, e.g. the K-FAC like approximation.
>
> Thank you for this remark.  We did not develop this aspect but essentially Newton method is affine invariant: it is possible to get normalized variables through an affine transformation. Variance matrix become identity and K-FAC like approximation is no longer needed. This point  would requires substantial space to develop (more than 2 pages).
>
> >2. I appreciate that the authors provide Section 1 to remind readers how to derive backpropagation using lagrange multipliers. That is very intuitive, introducing auxiluary variables and lagrange multipliers does not change the final result. However, when moving to section 2, it is completely unclear where the Newton's update comes from.
>
> We do not think that the above statement is correct: the Lagrange multipliers method is used to derive the gradient and the adjoints/multipliers are a necessary part of the procedure.  The description of how the “final result” does not change with the introduction of auxiliary variables is confusing in our opinion.
>
> >*	Notation and nomenclature ...they use confusing wording: "positive symmetric matrices" instead of "positive definite matrices", "adjoint state vectors" instead of "lagrange multipliers".
>
> First, if there are any unconventional notations, we will of course fix such issues.
> Second, adjoint state-vector and Lagrange multipliers are the same objects and both appellations are used in different fields (Control theory, optimization, … etc.).
> Finally, symmetric, and definite refer to different properties, hence the above comment is rather not accurate. The words we used have the appropriate meaning.
>
> >3.	Finally, the "Results" section: by just looking at the plots, it seems that the proposed method is worse than most of competitors. Even worse, there is no comment or interpretation to make sense of the results.
>
> The results are a standard comparison of validation and training errors to showcase that the Exact Stochastic Newton method, is properly working and not diverging.  We are also performing better than some of the best standard methods. The results could be naturally improved with finetuning.

---

### Official Review · Reviewer_3eEq · 2021-11-03

**Correctness:** 4
**Technical Novelty And Significance:** 2
**Empirical Novelty And Significance:** 2
**Recommendation:** 6
**Confidence:** 2

**Main Review:**

The work extends the 1st order results in (LeCun et al., 1988) to include 2nd order information for feed forward neural nets, and experimentally compares the new 2nd order methods with other neural net training techniques. The Sifrian formulation is interesting and provides new neural net training methods. There is a potential issue for the Sifrian to be hard to solve.
However, the paper requires the existence of a peculiar regularizer, but does not discuss any statistical properties of this regularizer.

**Summary Of The Paper:**

The paper considers feed forward networks as a base model, build a second-order Lagrangian which it calls Sifrian,
and provides a closed-form formula for the exact stochastic Newton direction under some monotonicity and regularization conditions. It proposes a convexity
correction to escape saddle points, and it reconsiders the intrinsic stochasticity of
the online learning process to improve upon the formulas.

**Summary Of The Review:**

The work extends the 1st order results in (LeCun et al., 1988) to include 2nd order information for feed forward neural nets, and experimentally compares the new 2nd order methods with other neural net training techniques. The Sifrian formulation is interesting and provides new neural net training methods.  There is a potential issue for the Sifrian to be hard to solve. However, the paper requires the existence of a peculiar regularizer, but does not discuss any statistical properties of this regularizer.

---

> ### Author Response · Authors · 2021-11-10
> **Statistical Effect of Regularization**
>
> The authors would like to thank the reviewer for his comments and feedback. We agree that several methods could be derived from the Sifrian equations and that solving the general case is challenging.
>
> The regularization we have used is meant to provide constraints on layers where no information is available: this is the case of all hidden layers since comparison with labels is only performed at the output. When regularization is not included, we get a “granular” solution that updates only the bias of the last layer (before the output) and hence is not particularly useful for the learning task. We will add in the appendix a paragraph to further elucidate the statistical effect of the regularization, which revolves essentially around keeping the norm of the state variable x small (relatively to the non-regularized case). Moreover, this type of regularization implicitly minimizes the norm of the gradient.
>
> The result we got in this work ultimately allows the leveraging of the Hessian information efficiently, exactly (without any truncations), and without divergence issues. That is the main novelty of this paper.

---

### Decision · Program_Chairs · 2022-01-20

**Decision:**

Reject

**Comment:**

This paper presents a second-order optimization algorithm for neural nets which extends LeCun's classic Lagrangian framework. The paper derives a method for computing the exact Newton step for a single training example for a multilayer perceptron. It then describes approximations that can be used to extend the method to more examples.

The authors claim to have spotted factual errors in the reviews. However, I've looked into the issues, and I find myself agreeing with the reviewers on each of those points (or, if there are misunderstandings, they result from a lack of clarity in the paper rather than insufficient scientific computing background on the part of the reviewers).

The authors claim to have solved a longstanding problem by giving an efficient method for calculating the stochastic Newton step (for a single training example). However, it's not clear this is very useful; as a reviewer points out, estimating the curvature with a single example can't give a very accurate estimate. Once the method is extended to batches, more approximations are required. I also agree with the reviewers that the later parts of the methods section appear a bit rushed.

As the reviewers point out, in the experimental comparisons, the proposed method seems to underperform SGD with momentum even in terms of epochs, which is the setting where second-order methods usually shine. Other second-order optimizers (e.g. K-FAC) have been shown to outperform first-order methods in terms of both wall clock time and epochs, so epochwise improvement seems like the minimum bar for a second-order optimization paper.